# Radiolytically reworked Archean organic matter in a habitable deep ancient high-temperature brine

Devan M. Nisson [1] ✉, Clifford C. Walters [2], Martha L. Chacón-Patiño[3], Chad R. Weisbrod[3], Thomas L. Kieft[4], Barbara Sherwood Lollar[5,6], Oliver Warr [7], Julio Castillo[8], Scott M. Perl [9], Errol D. Cason [10], Barry M. Freifeld[11] & Tullis C. Onstott[1]

Investigations of abiotic and biotic contributions to dissolved organic carbon (DOC) are required to constrain microbial habitability in continental subsurface fluids. Here we investigate a large (101–283 mg C/L) DOC pool in an ancient (>1Ga), high temperature (45–55 °C), low biomass ($10^2$–$10^4$ cells/mL), and deep (3.2 km) brine from an uranium-enriched South African gold mine. Excitation-emission matrices (EEMs), negative electrospray ionization (−ESI) 21 tesla Fourier-transform ion cyclotron resonance mass spectrometry (FT-ICR MS), and amino acid analyses suggest the brine DOC is primarily radiolytically oxidized kerogen-rich shales or reefs, methane and ethane, with trace amounts of $C_3$–$C_6$ hydrocarbons and organic sulfides. $\delta^2H$ and $\delta^{13}C$ of $C_1$–$C_3$ hydrocarbons are consistent with abiotic origins. These findings suggest water-rock processes control redox and C cycling, helping support a meagre, slow biosphere over geologic time. A radiolytic-driven, habitable brine may signal similar settings are good targets in the search for life beyond Earth.

The production of dissolved organic carbon (DOC) species in subsurface fluids is typically described as either biotic or abiotic in origin. In all cases, however, the combined effects of production coupled with subsequent alteration by both microbial and abiotic water-rock processes, as well as thermal alteration and burial, must be evaluated to understand the system and the rate balance of biotic-abiotic processes on the observations[1–5]. The relative rates of biotic and abiotic processes controlling carbon cycling may vary significantly as a function of the local environment[6]. Shallower groundwaters in both sedimentary and crystalline rock aquifers where meteoric waters have penetrated are shown to contain a greater diversity of organic molecules

relative to fracture fluid systems that have experienced greater periods of isolation since their last episode of meteoric fluid infiltration[7,8]. This discrepancy includes larger concentrations of aromatic humic and fulvic acids, derived from current or recent mixing with pools of surface-photosynthate delivered to these shallower systems via groundwater recharge[7,8]. These hydrogeologic connections can be modern, or may reflect geologic processes, as in particular, paleo-recharge from past glacial cycles has been invoked as a global mechanism for enhanced penetration of surface waters into the near subsurface, sometimes to kilometers depth[9,10]. Subsurface systems can also contain much older biotic input if leaching occurs from proximal

[1]Department of Geosciences, Princeton University, Princeton, NJ 08540, USA. [2]Bureau of Economic Geology, University of Texas, Austin, TX 78758, USA. [3]National High Magnetic Field Laboratory, Tallahassee, FL 32310, USA. [4]Department of Biology, New Mexico Institute of Mining and Technology, Socorro, NM 87801, USA. [5]Department of Earth Sciences, University of Toronto, Toronto, ON M5S 3B1, Canada. [6]Institut de Physique du Globe de Paris (IPGP), Université Paris Cité, 1 rue Jussieu, 75005 Paris, France. [7]Department of Earth Sciences, University of Ottawa, Ottawa, ON K1N 6N5, Canada. [8]Department of Microbiology and Biochemistry, University of the Free State, Bloemfontein 9300, South Africa. [9]NASA Jet Propulsion Laboratory, California Institute of Technology, Pasadena, CA 91109, USA. [10]Department of Animal Sciences, University of the Free State, Bloemfontein 9300, South Africa. [11]Lawrence Berkeley National Laboratory, Berkeley, CA 94720, USA. ✉e-mail: dnisson@princeton.edu

kerogen/bitumen rich ore deposits, such as the case with coal beds and oil field systems[11]. Due to the higher DOC of these systems, they typically contain high microbial biomass ($\geq 10^5$ cells/mL) with favorability of aerobic, heterotrophic strategies, in part also due to mixing with more current/recent microbe-rich oxic surface fluids[8,12,13]. In the case of some oil field brines where a hiatus in fluid recharge can occur, these systems can evolve and accumulate high molecular weight, more recalcitrant species in addition to high levels of biogenically produced methane and lipids as the leftovers of thermogenic breakdown and anaerobic microbial reworking[14,15].

At even greater depths (>2 km), isolation from surface photosynthate can persist in fractured rock aquifers for considerably longer periods of time (residence times ~Ma-Ga timescales)[6,16–20], resulting in dominantly oligotrophic highly saline fluid environments. Depending on the hydrogeochemical setting, these groundwaters can harbor large (DOC) concentrations[21,22], but these fluids tend to have lower diversity in organic species, with dominance of lower molecular weight organic compounds such as abundant $n$-alkanes and aliphatic acids[13,20,23]. Depending on the geologic history of the system, hydrocarbons in these groundwaters can be biotic in origin, from in-situ microbial production or the products of thermal degradation[24–26] and/or abiotically contributed from a variety of water-rock reactions related to radiolysis, serpentinization and Fischer-Tropsch type synthesis reactions[3,20,27]. Deep fracture fluids with low water/rock ratios and Ma-Ga residence times can generate and accumulate hydrogen, abiogenic methane and $C_{2+}$ hydrocarbons produced from these water-rock reactions, which can serve as precursors in the production pathways for various other abiotic and biogeochemical processes[1–3,5,20,28–30]. Where the continental crust has reached biologically habitable temperatures (<150 °C), they can also support biomass (typically low $\leq 10^4$ cells/mL) that may also contribute organic carbon that is recycled within the community[31]. The seeming paradox of extremely low biomass, in the presence of abundant organic substrates, suggests that environmental constraints in these deep systems are limiting the size and structure of microbial communities utilizing these biogenic and/or abiogenic carbon sources[19,26,28].

A constraint frequently reported in subsurface fluids in regions of low permeability is that of increased salinity, commonly the result of long-occurring water-rock interactions and isolation from modern fluid recharge[22,32–34]. Previous microbial characterization of <2 km deep, mesophilic (27–37 °C) and brackish (4–11 g/L) fracture fluids in the Witwatersrand Basin found microbial communities reliant on primary production of methane by a small fraction of lithoautotrophic members[23–25,35]. Alternatively, in 1.7 -Ga hypersaline brines (190–220 g/L) from a 2.4 km deep, mesophilic (25–26 °C), and radiogenic (~2 ppm $U^{238}$) fracture system at Kidd Creek Observatory in the Canadian Shield, predominantly autotrophic sulfate-reducing and alkane-oxidizing microbes were identified in selective most probable number (MPN) enrichments[19]. While MPN approaches have a high potential for false negatives, it is notable that no methanogenic activity was identified[19], although a clumped methane isotope study later identified a small effect on clumped isotopes attributed to anaerobic methane oxidizers[34]. The high dissolved organic carbon load in these low biomass environments (e.g., 4.92 mg C/L, 10–$10^2$ cells/mL Witwatersrand Basin[22,35]; 28.8–60.0 mg C/L, $10^3$–$10^4$ cells/mL Kidd Creek[19,20]) suggests carbon compounds derived from ancient organic deposits or produced through abiotic processes may help sustain mesophilic communities in hypersaline subsurface environments in contrast to primary production-dependent communities in younger, brackish environments where microbes are not osmotically challenged[36].

To date, however, the intersection between abiotic and microbial processes in these deep and isolated settings have only been characterized in a limited number of sites. Consequently, little is known on the intersection between abiotic processes and environmental conditions with respect to habitability in the deep crust. Here, we extend these investigations to a deep subsurface brine system in Moab Khotsong, South Africa, which is to date the only other site where subsurface fluids with Ga residence times have been found[37]. Previous work on two brine sites from Moab Khotsong revealed higher temperature (45–55 °C) and salinity (215–246 g/L), than the aforementioned fracture fluid environments[38]. The brines of Moab Khotsong additionally experience increased exposure to ionizing radiation from radionuclide decay of local host rock species ($\leq 1.1$ ppm U, $\leq 3.2$ ppm Th, $\leq 0.50\%$ K)[38] and from the proximal uranium-enriched Vaal Reef (~2800 ppm U)[39], with previous radiogenic noble gas- and $^{36}Cl/Cl$-based estimates suggesting a minimum contribution of at least 11–20 ppm and potentially up to 100 ppm of U from the reef[37,38,40]. The distinct abiotic nature of the two brines was further highlighted through geochemical characterization of a third, lower salinity (0.712 g/L) and temperature (26 °C) dolomitic fluid within Moab Khotsong that displayed higher biomass ($10^6$ cells/mL) relative to the brines ($10^2$–$10^3$ cells/mL)[38].

With the previous characterization of Moab Khotsong geochemistry relative to that for global Precambrian Shield brines, including the ancient brines of Kidd Creek, we are now in a position to better understand the role of abiotic-biotic carbon cycling in these deeper fluid systems that have experienced extended subsurface residence times and greater degrees of isolation from meteoric fluid mixing[34]. The overall objective of this study is to determine organic carbon origin and availability for microorganisms persisting in a long-isolated (1.2 Ga[37]) deep brine system. Specific objectives are (1) to identify the constituent components and relative concentrations contributing to the DOC pool in the Moab Khotsong brine, (2) to determine how secondary abiotic or biotic alteration may affect the current organic pool, including contributions from dissolved inorganic carbon (DIC). The production and sustainability of organic carbon is of great interest not only in the terrestrial subsurface, but also in potentially long-isolated, highly saline, and extreme temperature subsurface fluids in other planetary settings, as the presence of organics is often used as a proxy for habitable potential[41]. Additionally, such planetary environments often experience high levels of ionizing radiation[42–45]. The combination of these conditions in the Moab Khotsong brine environment thus makes it a valuable analog site for understanding the potential for microbial life to persist under extreme conditions isolated from a planet's surface on long time scales.

## Results

Fluids considered in this study were sampled from Moab Khotsong gold and uranium mine, located in the 2.10–3.00 Ga Witwatersrand Basin of South Africa. Specifically, two brine systems were sampled from boreholes located at 2.9 km and 3.1 km depth within Archean age (2.90 Ga) quartzites of the West Rand, representing the brines from the 95- and 101-levels, respectively. Additional analyses were performed on paleo-meteoric fluid from a karstic aquifer at 1.2 km depth within the 2.20–2.50 Ga Transvaal Dolomites (referred to as the 1200-level) and on service water circulated throughout the mine. Analyses of these two lower salinity fluids are presented to determine the potential of contamination and fluid mixing with the brines.

### Old biotic contributions to brine DOC

Overall concentrations of DOC and DIC pools were determined for the Moab Khotsong brines (95- and 101-levels) and dolomite fluid (1200-level), and these values accompanied by isotopic measurement of $\delta^{13}C$ and $\Delta^{14}C$ were used to help determine carbon source and potential utilization. Large differences between levels were found in DOC and DIC concentrations (Fig. 1b).

DIC of the 1200-level dolomite aquifer (83.7 mg C/L) was greater relative to the brines (3.2 mg C/L (95-level) and 12.9 mg C/L (101-level)). This trend was reversed for DOC composition, in which the brines contained much greater DOC concentrations (101 mg C/L (95-level)

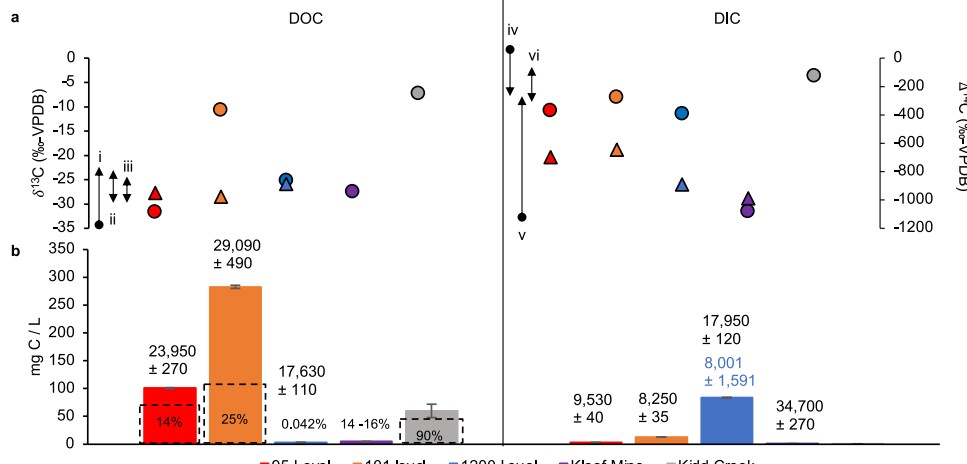

**Fig. 1 | Dissolved organic and inorganic carbon (DOC/DIC) and δ¹³C, Δ¹⁴C signatures. a** $\delta^{13}C$ (circles) and $\Delta^{14}C$ measurements (triangles), relative to Vienna Pee Dee Belemnite (V-PDB), for DOC and DIC pools from (**b**). Reproducibility of $\delta^{13}C$ is ±0.3‰ for Moab Khotsong and Kloof samples, and ±2.5‰ for Kidd Creek. $\delta^{13}C_{DOC}$ ranges include: (i) reported values for DOC of Witwatersrand Basin fracture waters (full range -24 to -57‰[23,35]), (ii) extractable organic carbon from Vaal Reef[55], (iii) extractable organic carbon from the Kimberley Shales[102]. $\delta^{13}C_{DIC}$ ranges include: (iv) acetogenesis or methanogenesis (full range of −7 to 12‰[22,103]), (v) reported values for DIC of Witwatersrand Basin fracture waters (full range of -12 to -43‰[23,35]) (vi) calcites from Moab Khotsong[38]. $\delta^{13}C_{DIC}$ of 0.6‰ for Transvaal Dolomites[104] is out of range. **b** Quantity of DOC and DIC in mg C/L of the Moab Khotsong fracture fluids compared to Kloof Mine[35] and Kidd Creek brine[20] (Kidd Creek DIC (0.7 mg C/L) is smaller than other plotted samples). Error bars represented as relative standard deviation (% RSD) and are ±1% for Moab Khotsong and Kloof Mine samples and ±20% Kidd Creek, and in some cases, are smaller than the symbols. Percentages in dashed boxes indicate the percentage of low molecular weight organic acid (acetate + formate + propionate + oxalate) comprising the DOC pool. Apparent ages based on $\Delta^{14}C$ from (**a**) are shown in years above each carbon pool ± relative error. Blue ages for DIC of the 1200-level include correction for carbonate dissolution upon aquifer recharge.

and 283 mg C/L (101-level) vs. 3.4 mg C/L (1200-level) along with greater proportions of low molecular weight organic acids, primarily acetate and formate. With regards to $\delta^{13}C$ composition, values for DIC were slightly more enriched for the brines (−10.8‰ (95-level) and −8.1‰ (101-level)) compared to the dolomite (−11.4‰) (Fig. 1a). The same pattern was found for the 101 brine DOC pool, with a value of −10.7‰ versus −25.1‰ for the dolomite. The 95-level brine DOC pool was lighter than these two fluids with a signature of −31.6‰. $\Delta^{14}C$ values ranged from −974.01 to −948.61‰ and −697.19 to −644.96‰ for the brine DOC and DIC pools, respectively, with associated apparent ages estimated from 23,950 to 29,090 and 8250 to 9530 years. Calculations to distinguish the source of $^{14}C$ introduction to the system are considered in detail in Supplementary Information. The 1200-level displayed a more positive $\Delta^{14}C$ for DOC (−888.54‰) but a more negative value for $\Delta^{14}C$ of DIC (−893.68‰). Not shown in Fig. 1 are the low concentrations of DOC found in the service water (5.4 mg C/L; −54.8‰ $\delta^{13}C$) with higher DIC (17.6 mg C/L; −32.6‰ $\delta^{13}C$).

Excitation-emission matrices (EEMs) were collected for the Moab Khotsong fluids in addition to an extract from the proximal and kerogen-enriched Vaal Reef to see if this might be a potential source of organics for the brines. This technique utilizes a dual scanning fluorometer to determine fluorescence for components in a DOC sample across a broad range of emission and excitation wavelengths. This can be visualized as a spectrum of fluorescence intensities, and peak intensity regions can be compared to characteristic peaks of previously identified DOC molecular groups with a fluorescent signature (e.g., humic-like compounds, aromatic amino acids, contaminant dyes[35,46]). The EEMs spectra for the brines displayed broad similarity in identified peaks and peak intensity (Fig. 2), supported by an alternative analytical process (UV-Vis absorptivity, Supplementary Information Fig. S1).

Exclusion of the high intensity fluorescein peaks for the 95-level sample reveals a likely combination of several lower intensity peaks, including: tyrosine-like protein B peaks ($\lambda_{ex}$ ~ 275/$\lambda_{em}$ ~ 305), tryptophan-like protein T peaks ($\lambda_{ex}$ ~ 275/$\lambda_{em}$ ~ 340), and humic-like A and N peaks ($\lambda_{ex}$ ~ 260-270/$\lambda_{em}$ ~ 380-460 and $\lambda_{ex}$ ~ 280/$\lambda_{em}$ ~ 370,

respectively). The 101-level spectrum also showed tyrosine-like protein B and humic-like A peaks, but displayed an autochthonous microbial DOC M peak ($\lambda_{ex}$ ~ 300/$\lambda_{em}$ ~ 400) not seen for the 95-level brine. The 1200-level dolomite fluid displayed humic-like A and N peaks in addition to a tryptophan-like protein T peak and fluorescence in the M peak region, while the Vaal Reef extract was a much simpler spectrum with one predominant peak most similar to a humic-like A peak.

Fracture fluids were additionally characterized by negative ion ESI 21 tesla FT-ICR MS. This technique provides the highest mass accuracy and resolution, enabling the detection of tens of thousands of peaks within a given DOC sample, with subsequent molecular formula assignment for complex mixtures such as fossil fuels, natural organic matter, and samples of environmental interest[47]. In this study, the majority of the detected peaks in each sample were assigned a molecular formula, which enabled the determination of double bond equivalent (DBE) values, carbon numbers, and atomic ratios such as H/ C and O/C. The percentage of unidentified peaks observed was lowest for the 1200-level (11%), increasingly higher for the 101 (15%) and 95 (20%) level brines, and greatest at 24% for the Vaal Reef DOC extract. Most of the non-assigned peaks (no-hit) correspond to random contaminant species whose composition does not fit into extended, continuous Kendrick series and, therefore, cannot be attributed as dissolved organic matter[48]. The assigned molecular formulas were then visualized in Van Krevelen diagrams of atomic ratios, in this case as H/C vs O/C[49]. The 95-level sample molecular profile was dominated by fluorescein at 98% relative abundance (based on identified peak $C_{20}H_{12}O_5$ with DBE of 15 and m/z 331.06). The highest contaminating classes from extraction, based on an MQ $H_2O$ procedural blank, were $O_2$ and $O_3S_1$, but these were less abundant than the brines, based on low overall measured DOC of MQ $H_2O$ (-1 mg C/L) (Supplementary Information Fig. S3 and Fig. S5). This was also the case for the low DOC service water pool with regards to $O_{4+}$ and $O_5S_1$ molecular classes. Contamination from organic drilling additives appeared to be low across abundant molecular classes shared with the brines (Supplementary Information Fig. S3 and Fig. S5). The highest abundance of potential contamination was seen across $O_5S_1$ to $O_9S_1$ classes for the

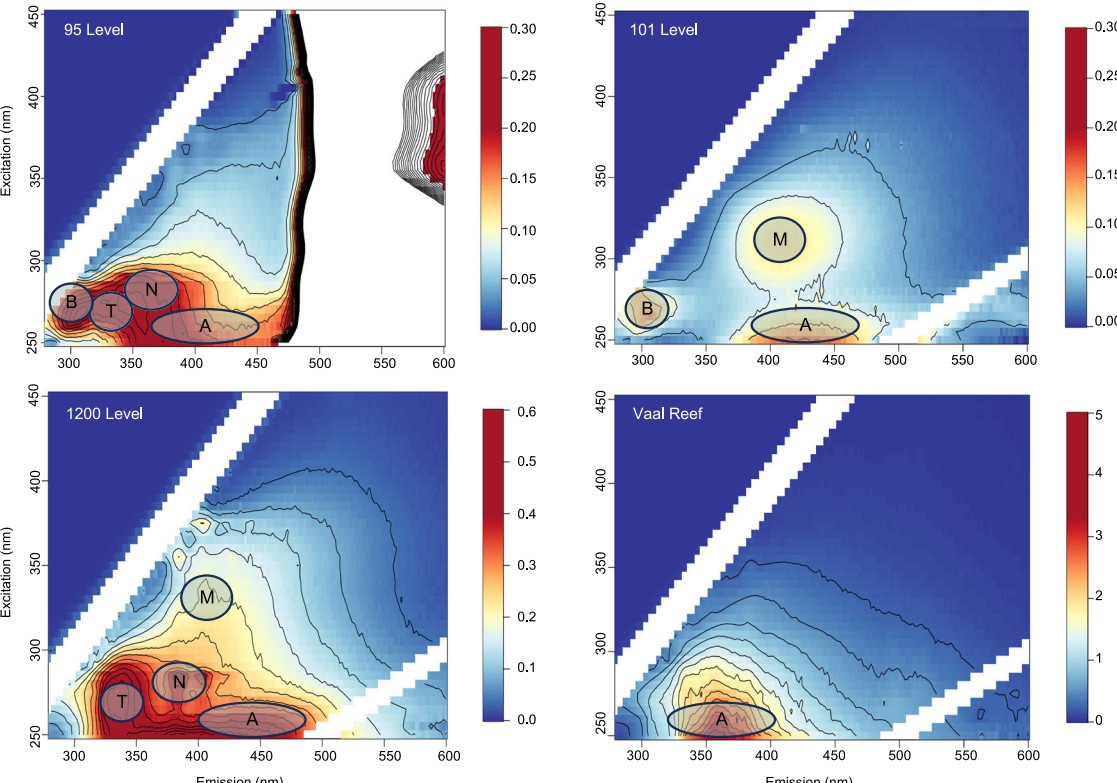

**Fig. 2 | Excitation emission matrices (EEMs).** EEMs shown are for the 95-level brine, 101-level brine, 1200-level dolomite fluid, and Vaal Reef extract. Scales display fluorescence intensity values with contours at 0, 0.02, and 5. Diagonal features represent first and second order Raman scatter. Peak regions are identified by corresponding lettered circles[35,46]: A,N – humic-like; B – tyrosine protein-like; T – tryptophan protein-like; M – autochthonous microbial DOC. The large white area from ~480 to 600 nm emission in the 95-level spectrum is due to the presence of fluorescein dye in this sample and the scale has been adjusted to exclude the high fluorescence of this region. The full spectra for the 95-level 2018 to 2020 spectra with both this excluded vs. included fluorescein signal can be seen in Supplementary Information Fig. S2.

AMC EzyCore, but the overall abundance of these same classes was low for the brine.

Not only were DOC concentrations as well as $\Delta^{14}C$ similar for the two Moab Khotsong brines, but so too were the overall distributions of organic species as determined by FT-ICR MS (Fig. 3; Supplementary Information Fig. S4 and Fig. S6; Supplementary Data 1). Both displayed a broad O/C range (~0 to 0.9) and had a greater degree of aliphatic carbon relative to aromatics and condensed aromatic structures according to their H/C distribution (~0.4 to 2.3). The Vaal Reef aqueous extract also displayed a similar overall molecular distribution to the brines, with low levels of aromaticity in favor of more aliphatic compounds, but with fewer molecules of O/C > 0.6. The 1200-level fluid, the sample with the lowest DOC concentration, had a much more limited and overall higher H/C range (~1.0 to 2.0) and fewer molecules in high O/C regions (~0.1 to 0.8), with clear enrichment in zones similar to lipid and protein-like molecular compositions. Most of the Van Krevelen diagrams were distinct from the terrestrial-plant-derived Suwanee River fulvic acid standard; however, the service water most closely resembled that standard, showing H/C and O/C enrichment in tannin-like regions (Supplementary Information Fig. S3).

Sorting peak identifications by heteroatom class revealed that the majority of molecules fit into either $O_x$, $N_1O_x$, or $S_1O_x$ molecular type (Fig. 4). The 1200-level fluid as well as the 95 and 101-level brines also contained several species belonging to $^{35}Cl$-, $^{37}Cl$-, or Na-containing heteroatom classes that were not shared by the other samples; however, these species displayed lower overall relative abundance and are not included among the comparison of more abundant classes presented here (Supplementary Information Fig. S4). The most abundant

$O_x$ species in the brines included those in $O_4$ (DBE 1–15), $O_5$ (DBE 1–14), and $O_6$ classes (DBE 1–15); the 95-level brine also contained high abundance of $O_2$ and $O_3$ species. Within the $O_4$–$O_6$ classes, abundance for molecules within a particular DBE value peaked at derivatives of $C_{18}H_{36-(DBE+2)}O_x$ for DBE < 5, and $C_{20}H_{40-(DBE+2)}O_x$ for DBE > 5 (where n is 4, 5, or 6). Generally, abundances were highest for molecules with DBE of 2, 3, and 6, consistent with biologically derived fatty acids (Supplementary Information Fig. S6a–c). Although both brine samples followed these trends, the 101-level brine often displayed higher abundance of lower carbon number species within a DBE value relative to the 95-level, which may be due to the higher DOC content of this sample. The other most abundant organic class in the brines included $O_3S_1$ species. Among this class, most of the molecules identified in the brines showed a DBE of 4 and peaked in the $C_{16}$-$C_{19}$ (Supplementary Information Fig. S6d). The 101-level displayed a greater relative abundance of species in the $S_1O_x$ class indicating an enriched S organics pool relative to the 95-level. Both brine samples had low relative abundance of $N_1O_x$ species.

Among the brine FT-ICR MS distributions, the 1200-level displayed higher relative abundance for larger species in each heteroatom class. The 101-level brine and the 95-level brine both showed higher abundances shifted towards lighter species in each class (Fig. 4). Additionally, the 1200-level contained more species in the $N_1O_x$ class than either of the brines, indicating a higher overall relative N content in the DOC for that fluid. The Vaal Reef sample showed greatest relative abundance for similar molecular classes as the brines, in particular the $O_4$-$O_6$ and $O_3S_1$ classes. Within these $O_x$ classes, the Vaal Reef showed similar abundance peak distributions, but at lighter compounds for each DBE value (Supplementary

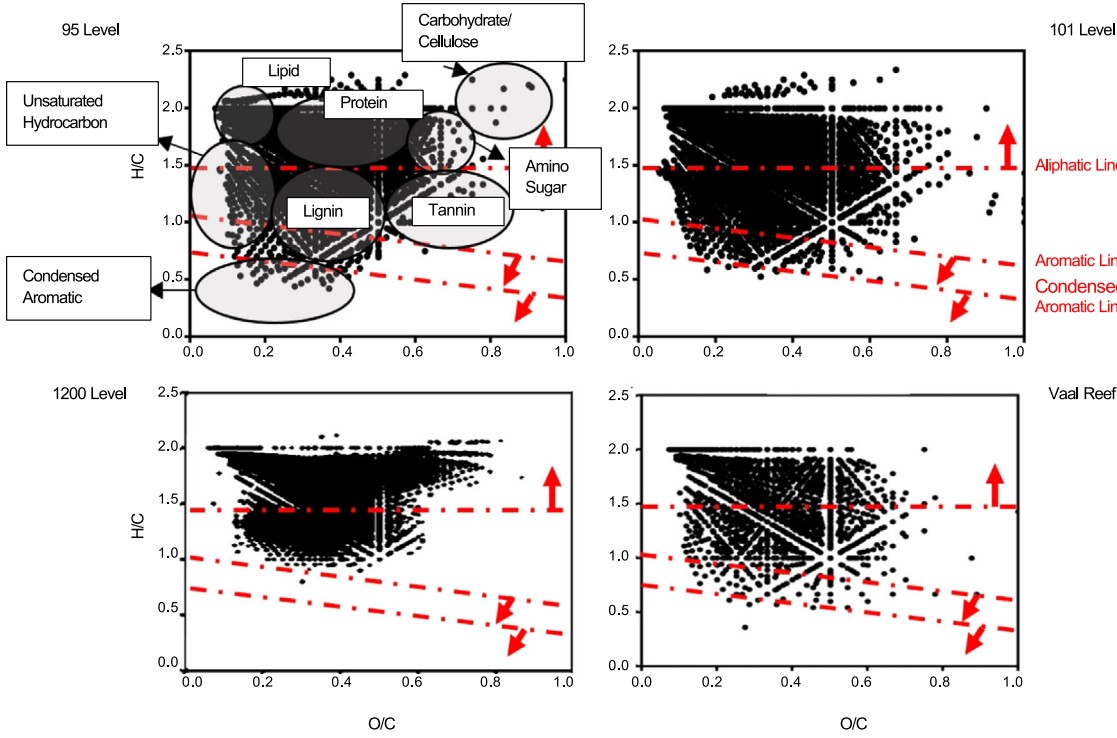

**Fig. 3 | Van Krevelen diagrams from negative electrospray ionization 21 tesla Fourier-transform ion cyclotron resonance mass spectrometry (FT-ICR MS).** Samples shown include the 95 and 101-level brines compared to the 1200-level dolomite fluid, and an extract from the Vaal Reef. Dashed red lines and arrows indicate regions of increasing (1) aliphatic, (2) aromatic, and (3) condensed aromaticity of molecules. Circles included on the 95-level diagram represent typical H/C vs. O/C regions for broad organics classes[105].

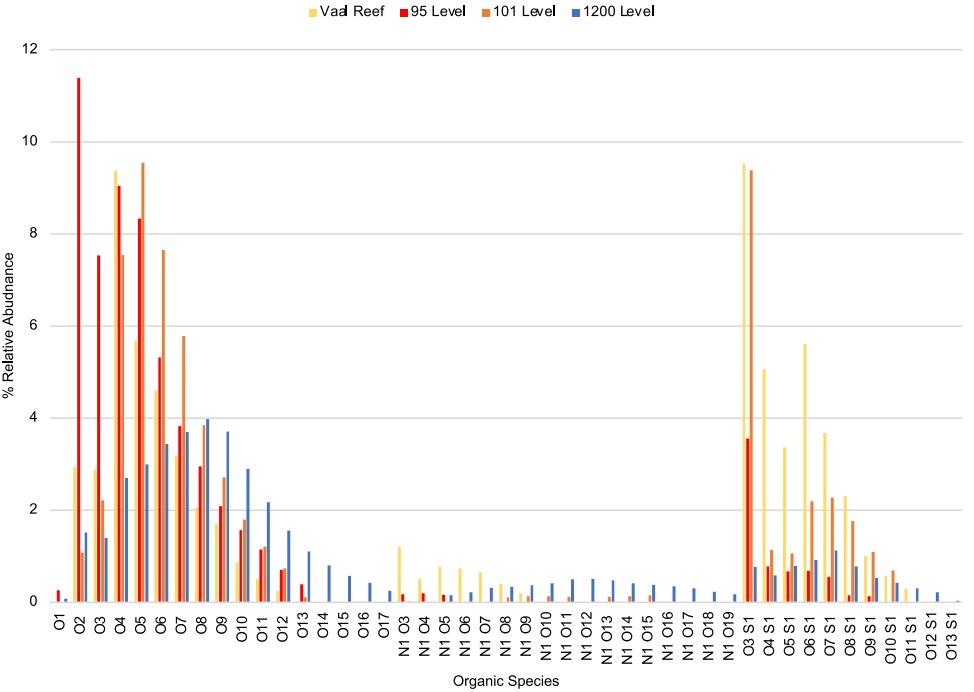

**Fig. 4 | Percent relative abundance of organic species.** Major classes $O_x$, $N_1O_x$, and $S_1O_x$ obtained through negative ion ESI 21 tesla FT-ICR MS for 95 and 101-level brines compared to the 1200-level dolomite fluid and an extract of the kerogen-rich Vaal Reef.

Information Fig. S6a–c). Like the 1200-level, the Vaal Reef showed a greater abundance of $N_1Ox$ species, although this distribution also favored lighter masses and lower DBE compounds relative to the 1200-level. The reef also displayed high abundance of species across the $S_1Ox$ class, with highest abundance at $S_1O_3$ like the brines. A closer look at the $S_1O_3$ species showed very similar distribution of highest abundance for DBE of 4, $C_{16}$-$C_{19}$ molecules (Supplementary Information Fig. S6d).

**Table 1 | D/L enantiomeric ratios and concentration of aspartic acid for hydrolyzed fracture fluid samples**

| Sample[a] | Conc. (mg/L) (super)[b] | D/L-Asp (super) | Conc. (mg/L) (pellet)[b] | D/L-Asp (pellet) | Estimated Cell Density (cells/mL) | Estimated % of DOC that is Amino Acids |
|---|---|---|---|---|---|---|
| 95-level Unf. | D: $1.3 \cdot 10^{-8}$ L: $2.1 \cdot 10^{-8}$ | 0.43–0.62 | D: $2.1 \cdot 10^{-11}$ L: $3.3 \cdot 10^{-10}$ | 0.06–0.07 | $10^{4.5}$ | $1.13 \cdot 10^{-7}$ |
| 95-level Fil. | D: $8.6 \cdot 10^{-9}$ L: $1.9 \cdot 10^{-8}$ | 0.53–0.55 | D: $3.3 \cdot 10^{-11}$ L: $1.1 \cdot 10^{-10}$ | 0.24–0.25 | $10^{4}$ | $9.13 \cdot 10^{-8}$ |
| 101-level Unf. | D: $1.5 \cdot 10^{-8}$ L: $2.9 \cdot 10^{-8}$ | 0.42–0.43 | D: $6.9 \cdot 10^{-10}$ L: $2.9 \cdot 10^{-8}$ | 0.36–0.38 | $10^{5}$ | $8.75 \cdot 10^{-8}$ |
| 1200-level Unf. | D: $7.4 \cdot 10^{-9}$ L: $3.6 \cdot 10^{-8}$ | 0.14–0.15 | D: $1.2 \cdot 10^{-9}$ L: $4.2 \cdot 10^{-8}$ | 0.01–0.03 | $10^{7}$ | $8.56 \cdot 10^{-6}$ |

[a]Unf. = unfiltered, Fil. = 0.2 µm filtered.
[b]Pellet samples represent the cellular or mineral pelleted pool, while the supernatant (super) represents the extracellular pool of amino acids.

### Current biotic contributions to brine DOC

To determine potential organics input from current biological activity, the concentration and associated ratio of right-handed to left-handed enantiomers (D/L ratio) were determined for the aspartic acid pool in each of the brines and the dolomite fluid. Aspartic acid is a relatively quick amino acid to racemize[50], allowing for the determination of low (D/L ratio < 1), high (D/L ≪ 1), or no (D/L = 1) current biotic contribution[50,51]. Analysis for the unfiltered brines revealed lower total concentrations of ($3.5 \cdot 10^{-10}$ to $3.0 \cdot 10^{-8}$ mg/L) vs. the supernatant fraction ($3.4 \cdot 10^{-8}$ to $4.4 \cdot 10^{-8}$ mg/L) (Table 1).

The pellet fractions also displayed lower D/L ratios (0.06 to 0.38) vs. the supernatant for each unfiltered brine sample (0.42 to 0.62), with the lower ratios typical of increased microbial repair of D-aspartic acid to L-aspartic acid in the pellet. The 95-level filtered sample showed a much higher pellet D/L ratio (0.24 to 0.25) than its unfiltered counterpart (0.06 to 0.07), indicative of extracellular aspartic acid. Enantiomeric ratios for all samples, regardless of the sample fraction, were <1, indicating no one sample had fully racemized. The 1200-level displayed more balanced aspartic acid concentrations between its pellet and supernatant fractions, with a much lower D/L ratio in the pellet (0.01 to 0.03). Amino acids made up only a very minor percentage of the DOC in each sample ($8.75 \cdot 10^{-8}$ to $8.56 \cdot 10^{-6}$%). Combined concentrations of aspartic acid were additionally used to estimate a cellular density of each sample. This calculation explicitly assumes that the aspartic acid concentrations are solely associated with intact cells. Resulting estimates of ~$10^4$ cells/mL were calculated for the 95-level and ~$10^5$ cells/mL for the 101-level, with a highest overall cellular density estimate of ~$10^7$ cells/mL for the 1200-level.

### Abiotic contributions to brine DOC

Qualitative gas chromatography mass spectrometry (GC-MS) analysis was employed on headspace gas of the 95-level brine to determine relative intensities of brine volatile organic carbon species distributed by retention time. The volatile organic carbon pool of the 95-level brine revealed primarily $C_{1-4}$ with minor amounts of $C_{3-5}$ $n$-alkanes, $C_5$-$C_7$ iso- and cycloalkanes, $C_6$ and $C_7$ aromatic hydrocarbons, and a suite of alkyl sulfide- and thiol-containing compounds (Fig. 5). Several parameters for these samples showed patterns previously identified at sites in the Witwatersrand Basin and around the world that have suggested a significant component of abiotic hydrocarbons. Specifically, these include the very low $CH_4/C_{2+}$ ratio (Supplementary Information Table S1), $\delta^2H$ and $\delta^{13}C$ for methane, and the relationship between these isotopic values for methane, ethane, and propane including increasingly heavy $\delta^2H$ with increasing hydrocarbon number (−412‰ $\delta^2H_{C1}$ to −198‰ $\delta^2H_{C3}$ 95-level and −361‰ $\delta^2H_{C1}$ to −208‰ $\delta^2H_{C2}$ 101-level) (Fig. 6)[3].

### Discussion

The results presented in this study reveal deep subsurface environments such as Moab Khotsong can contain a large, complex organic matter pool with evidence of both biotic and abiotic input. DOC for

both brines reveal the presence of $^{14}C$ produced in-situ via neutron capture (Fig. 1; Supplementary Information) and a greater abundance of both high O/C and low H/C compounds consistent with radiolytic reworking in uranium-rich systems (Fig. 3)[52,53]. The brine organic matter concentrations at both the 95 and 101-levels are two orders of magnitude higher, and the compositions are distinct from those of organic drilling additives, service water used in drilling and cleaning throughout the mine, as well as from fluid of a less saline, and lower temperature dolomite aquifer in the same mine (1200-level). Evidence for current microbial contribution was supported by organic signatures of autochthonous microbial DOC and aspartic acid D/L ratios <1 in the brines. These results suggest that the large organic matter pool in the brines may support a low biomass microbial community, despite environmental conditions being outside of those typically conducive for optimum habitability (i.e., temperatures (45 to 55 °C), hypersalinity (215 to 246 g/L TDS), and elevated levels of local radionuclide decay)[38]. In combination with the radiolytically sustained redox environment of this brine system including radiolytically generated $O_2$ and alternative electron acceptors such as $NO_3^{-}$[38,54], the Moab Khotsong brines are able to support habitability of a low biomass microbial system.

The large DOC content (101 to 283 mg C/L) in the Moab Khotsong brine pools, with only a small fraction attributable to low molecular weight organic acids (Fig. 1), suggests their primary source is the carbon-rich layers of the surrounding ancient host rocks. Negative $\delta^{13}C$ values (−31.6‰) of the 95-level brine DOC pool are consistent with an isotopic footprint similar to that of kerogen previously analyzed in the Jeppstown shales and Vaal Reef (−30 and −25‰, respectively)[55]. The more $^{13}C$-enriched $\delta^{13}C$ signature for the 101-level brine (−10.7‰) relative to the 95-level brine and Vaal Reef/Shale signatures may result, in part, from microbial utilization of this large "old" DOC, leaving an isotopically heavier residual pool (Fig. 1). This is consistent with the higher biomass for the 101 brine ($10^4$ cells/mL)[38]. The presence of an overall light $\delta^{13}C$ signature for these fluids, does not negate input of abiotically produced organics from inorganic precursors, but unlike Kidd Creek, the Moab Khotsong brines are not dominated by highly $^{13}C$-enriched acetate and formate[20]. Both higher rates of microbial utilization and potentially greater abiotic input may contribute to the heavier signature of the DOC pool at the 101-level. Further investigation into compound-specific isotopic signatures would be required to identify the presence of any $^{13}C$-enriched abiotic acetate and formate, whose signature (if present) is currently overshadowed by the larger DOC composition.

In contrast to the brine samples, the more depleted $\delta^{13}C$ (−25.1‰) of the 1200-level dolomite aquifer is consistent with Witwatersrand Basin aquifers that receive modern surface water recharge[23,35]. Similarly, depleted DOC $\delta^{13}C$ produced by microbial autotrophic fixation is apparent in younger Witwatersrand fracture fluid systems including the brine of Kloof Mine (−27.5‰)[35]. Heterotrophic activity is not reflected strongly in the low concentration DIC pools for the brines in

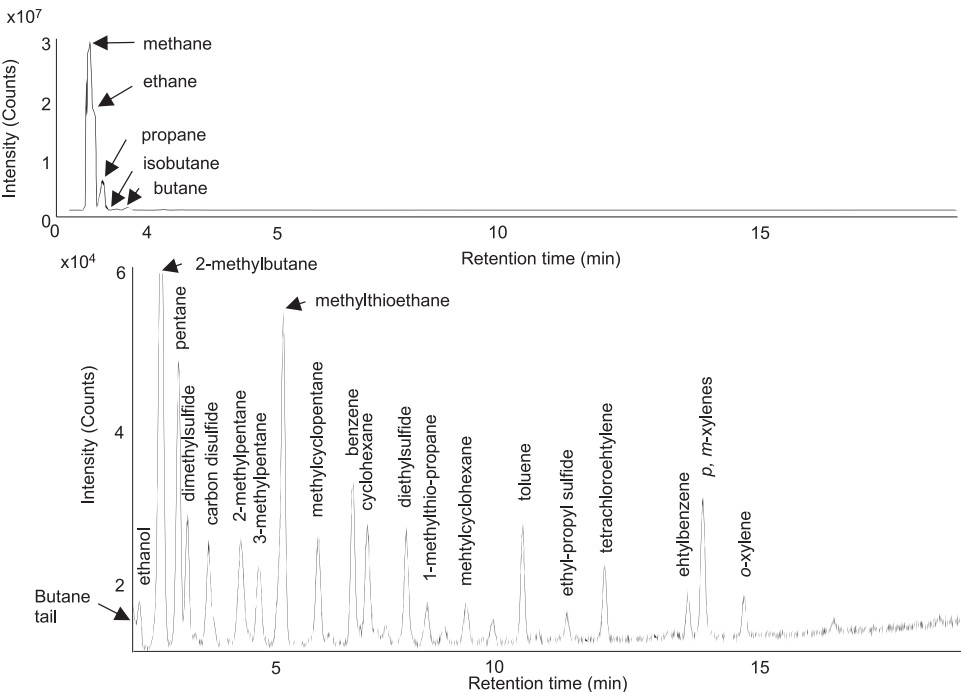

**Fig. 5 | Chromatographs of volatile organic species from headspace gas in the 95-level brine.** The top panel displays intensity of volatile species' peaks from 0 to $3 \cdot 10^7$ counts while the bottom panel displays species' peaks of lower intensity ($1.6 \cdot 10^4$ to $6.0 \cdot 10^4$ counts) over a 19.75-minute retention range.

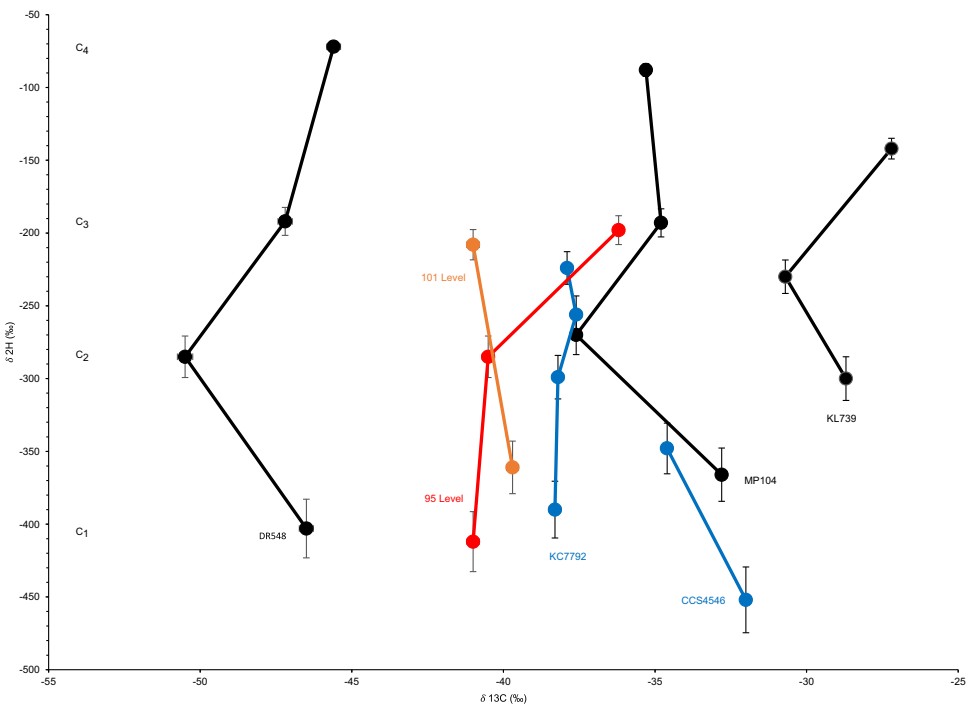

**Fig. 6 | Isotopic values (‰) for hydrogen and carbon of $C_{1-4}$ hydrocarbon gases.** Fluid systems including Moab Khotsong brines (95 (red) and 101 (green) levels), other fracture fluids previously sampled from the Witwatersrand Basin (Dreifontein −DR548, Mponeng−MP104, Kloof−KL739)[2], and fluids from the Canadian Shield (Copper Cliff−CCS4546)[2] including fluids from Kidd Creek Mine (KC7792) where residence times of over 1 Ga were identified (Sherwood Lollar et al.[1]). Error bars represent ±0.5‰ and ±5‰ relative to Vienna Pee Dee Belemnite (V-PDB) and Vienna Standard Mean Ocean Water (V-SMOW) for $\delta^{13}C$ and $\delta^2H$, respectively. Specific isotopic values and gas compositional data for $C_{1-4}$ hydrocarbons are included in Supplementary Material Table S1.

this study (3.2 to 12.9 mg C/L) with $\delta^{13}C$ (−10.9 to −8.1‰) compared to Witwatersrand DIC pools dominated by microbial respiration (∼−17‰)[23,25,56]. Considering the long isolation, low water/rock ratio, and low biomass of this system, the enriched $\delta^{13}C$ DIC here likely reflects interaction with surrounding calcites (−6 to −10‰)[38]. The DIC concentration of each Moab Khotsong brine is much less than the 1200-level (83.7 mg C / L) with the $\delta^{13}C$ of this site (−11.4‰) reflecting substantial dolomite dissolution. The $\Delta^{14}C$ values reported here suggest

negligible modern surface recharge and considerable in-situ generation of $^{14}C$. This can occur via neutron capture reactions relating to the high U (≥100 ppm) and Th (≥100 ppm) concentrations from the proximal Vaal Reef (Supplementary Information), a process that has been proposed in the literature[57], but only recently quantitatively evaluated for $^{14}C$ and related tracers in the continental subsurface[58,59].

In the EEMs spectra (Fig. 2), the bulk organic matter can be seen as fluorescence in the humic-like peak A region, which is similarly shared in the kerogen/bitumen rich Vaal Reef spectra, and may be indicative of kerogen-like marine or fluvially derived organic matter[60,61]. Similarities in abundant organics classes between the Vaal Reef and the brines are corroborated by negative ion ESI FT-ICR MS species identifications, with greatest abundances for $O_4$ to $O_6$ classes. The lower abundance of larger $O_x$ and $N_1O_x$ species in the reef sample relative to the brines is likely an artifact of taking an aqueous extract from the solid reef sample, with solubility of dichloromethane favoring extraction of lighter molecules[62]. The FT-ICR MS spectra suggest microbial reworking of the organic matter pool in the brines including a lower abundance of $N_1O_x$ compounds relative to the reef (in the reef ore zone, microbial activity is likely inhibited by high levels of radioactivity related to uranium decay. The host rocks are estimated to contain up to ~2800 ppm U)[39]. The abundance distribution of the brines relative to the 1200-level fluid may also suggest microbial activity, as this younger fluid displays a similar organics distribution to microbially re-worked organic pools in Witwatersrand Basin fluids (Fig. 3)[35]. A greater abundance of larger, more oxidized molecules in the 1200-level (followed by the 101 and 95-level brines, respectively) is indicative of greater heterotrophic utilization, and agrees with the trend of decreasing biomass for these fluids ($10^6$ cells/mL (1200-level) > $10^4$ cells/mL (101-level) > $10^2$ cells/mL (95-level)[38].

Greater geochemical evidence for increased microbial activity in the 101 and 1200-level fluids relative to the 95-level also appears in EEMs fluorescence in the peak M area, indicative of autochthonous microbial DOC[35,46]. This is further reflected in biomass estimates from low aspartic acid concentrations with accompanying D/L ratios <1 for these systems (Table 1). Cell density estimates based on aspartic acid concentration were 1–2 orders of magnitude greater than original Syto-9 or Raman-based cell counts for each sample[38]. This calculation implicitly assumes aspartic acid is associated with intact cells[50], and may result in an overestimate of cellular density if much of the aspartic acid is contributed by dead biomass and/or a dissolved amino acid fraction. This can be seen in aspartic-acid based cellular estimates of ~$10^4$ cells/mL for the filtered 95-level sample (Table 1), in which whole cells would be filtered out. This magnitude of difference between cellular estimates was within discrepancies previously reported for cell counts estimated from aspartic acid of Witwatersrand Basin fracture waters[50,51]. Although these analyses suggest microbial activity in the brines, the overall estimated biomass and relative abundance of active amino acid contribution are still very low and suggest relatively small overall contribution to DOC by in-situ microbiology.

This small contribution and manipulation by in-situ microbiology suggest abiotic water-rock interactions or thermogenic breakdown of old organic matter dominate DOC alteration and/or current production of organics in the ancient Moab Khotsong brines. Strong support for abiotic production of the volatile hydrocarbons (methane, ethane, propane) can be seen in the type of very low $CH_4$/$C_{2+}$ ratios and increasingly positive $\delta^2H$ and increasingly negative $\delta^{13}C$ signatures for $C_{2+}$ hydrocarbons relative to methane[1,2] (Figs. 5 and 6, Supplementary Information Table S1), previously seen at Kidd Creek and Copper Cliff South on the Canadian Shield, and sites in the Fennoscandian Shield and elsewhere on the Witwatersrand Basin, including several from the Carletonville mining district. Further support for abiotic production of the volatile hydrocarbons may be seen in the high relative abundance of $C_{2+}$ hydrocarbons versus methane and lack of $C_{6+}$ alkanes in the GC-MS profile for volatiles of

the 95-level brine (7.1 $CH_4$/$C_{2+}$, ~ 51% vol $C_1$-$C_4$) (Fig. 5; Supplementary Information Table S1), similar to the hydrocarbon dominated Kidd Creek volatiles (6.2 $CH_4$/$C_{2+}$,~88% vol $C_1$-$C_4$) measured by ref. 20. These signatures are distinct from methanogen-dominated fracture fluid systems of more mesophilic, less saline fracture waters where $C_1$ » $C_2$ + (typically ~1000 from ref. 29), or thermally mature oil field systems that contain long chain alkanes and polyaromatics among volatile species[3,29,63].

Microbial production pathways have been suggested for volatile sulfur-containing organics also seen in the GC-MS profile of the 95-level (Fig. 5) and Kidd Creek brine[29], but these distinctly involve methanethiol production[64], which has not yet been identified for the Moab Khotsong brines. Most non-microbial production pathways for these compounds include $H_2S$ as a reactant, with some options for its abiotic production including $H_2S$ generated by radiogenic $H_2$ + $SO_4^{2-}$ and reacting with in-situ organics[20,65], or thermochemical sulfate reduction and $H_2S$ generation from subsequent high temperature igneous intrusion[66]; both cases require the system to be long-isolated and to lack evidence for recent or current mixing with surface fluids, which are conditions met at Moab Khotsong[34,37,38]. These scenarios appear unlikely, however, given that previous measurements of $H_2S$ were below detection in the Moab Khotsong brines[38], and lack of evidence for igneous intrusion in the Witwatersrand Strata post 2.4 Ga[67]. Additionally, their production has been suggested to occur via radiolytic alteration of S-containing amino acids (e.g., cysteine and methionine), but these reactions have not yet been explored in natural groundwater systems and the specific quantification of these amino acids at Moab would require future positive ion ESI based MS detection and or targeted HPLC[68]. Although detailing the specific production pathway of these S-containing volatiles is outside the scope of the current study, the lack of microbial mechanisms for production of these species alongside their presence in other ancient brine systems indicate their potential abiotic origin.

Finally, the proposed radiolytic formation of the Moab Khotsong brines (≥0.3 Gy/year dosage[38]), alongside evidence of significant radiolytic oxidation and $^{14}C$ production in the organic pool (Supplementary Information) and low biomass indicated by cell counts[38] and aspartic acid densities (Table 1), suggests ancient planetary brines in radionuclide-rich regions, and regions that receive some organic input, may support habitable conditions over time. Potential targets for previously radiolytically-fueled extinct life include shallow subsurface aqueous environments of Mars exposed to galactic cosmic radiation (0.33 Gy/year dosage at 1 m depth for wet heterogenous terrain similar to Arabia Terra[43] and ~0.03 Gy/year dosage down to 1 m depth for dry homogenous terrain similar to Ares Vallis[44]). The low temperatures and likely low water activities of any current liquid brines in these shallow regions may support extant life only if it was more recently transported to near-surface regions[43]. There may also be greater possibility for extant life in deeper, ancient Martian brines fueled by radionuclide decay in the regolith ($4 \cdot 10^{-4}$ Gy/year at ≥ 3.6 m depth for dry homogenous terrain similar to Ares Vallis[44,69]), including populations of sulfate reducers (≤$10^6$ cells (kg rock)$^{-1}$ for Regolith brecca) supported by radiolytically produced substrates alone[70]. Other potential targets include putative deep subsurface brine reservoirs under the icy crust of dwarf planet Ceres[71,72], and the icy moons Europa[73] and Enceladus[74], where the decay of uranium, thorium, and potassium may contribute to the production of electron donors ($H_2$) and acceptors ($SO_4^{2-}$). These models, however, require further constraints on in-situ radionuclide concentrations and radiolysis vs. serpentinization rates for these icy worlds[75,76]. Besides receiving comparable doses of ionizing radiation, these systems are speculated to have in-situ contribution of abiotic organic species, including $CH_4$ produced via serpentinization, that may help sustain chemotrophic metabolisms utilizing abiotically contributed organic and/or inorganic energy sources[77]. The potential similarities between these prospective planetary sources and the

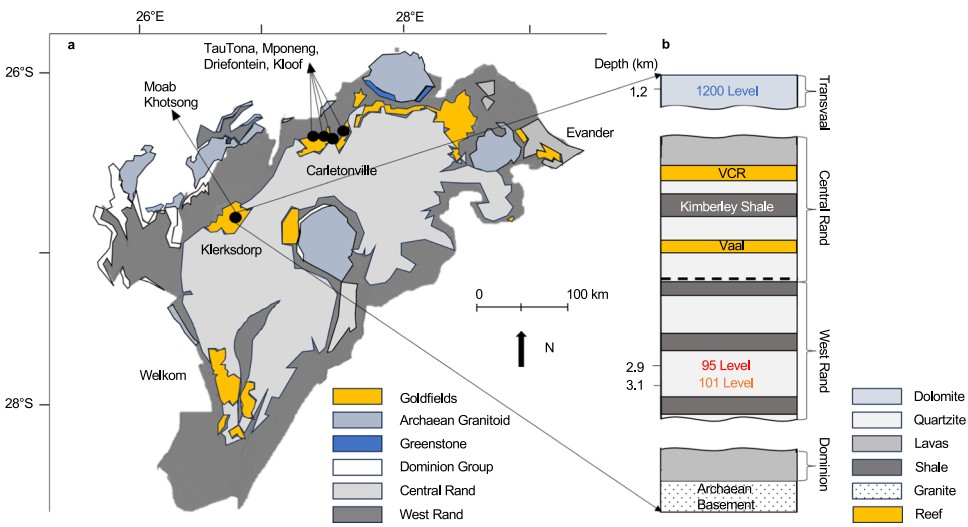

**Fig. 7 | Geographic and stratigraphic location of Moab Khotsong mine sampling sites. a** Location of the Moab Khotsong mine (along with Carltonville-based mines mentioned in this study) within the Witwatersrand Basin of South Africa (**b**) Stratigraphic location of facture fluid sampling sites in the Moab Khotsong mine showing the relationship of the boreholes in the Transvaal dolomite aquifer [1200-level; 1.2 km], and West Rand quartzites [95-level; 2.9 km and 101-level; 3.1 km], relative to major reef [VCR – Ventersdorp Contact Reef] and shale zones.

radiolytically-driven Moab Khotsong brines highlight this system as a novel terrestrial analog to evaluate habitability and biomarker detection in highly radioactive regions on other planetary bodies, where brines may have resided in the past (or present?). This work combined with future efforts to model microbial metabolism in planetary brines relative to Moab Khotsong will elucidate where best to search for signs of life in brine environments beyond Earth.

## Methods
### Geologic sampling site
Samples were collected from the Moab Khotsong gold and uranium mine (26.9792° S, 26.7815° E), in the Klerksdorp mining district of the Witwatersrand Basin in South Africa (Fig. 7).

Brine samples analyzed in this study were collected from the Witwatersrand Supergroup, which is divided into the 2.90 Ga to 2.79 Ga Central Rand Group (fluvial quartzites with minor conglomerates and shale), and the 2.99 Ga to 2.90 Ga West Rand Group (transitional marine-to-continental shale and quartzite)[78]. Deposition of this unit terminated at 2.79 Ga and the most recent localized metamorphism to occur in this section was at 2.02 Ga following the Vredefort meteorite impact[79], and potentially from 1 to 2 km depth during the uplift of the South African Plateau, if this event allowed for hydrothermal fluid circulation (90 to 70 Ma)[80]. Another, less saline fluid was sampled from the 2.65 to 2.40 Ga Chuniespoort Group in the Transvaal Dolomites, which is exposed at the surface and hosts a karstic freshwater aquifer. The Vaal Reef conglomerate is the major U-Au-rich ore zone mined in Moab Khotsong and lies in the center of the Central Rand Group[39]. This reef is also a site of highly refractory kerogen-rich units (the so called "carbon leaders") generally attributed to largely cyanobacteria-derived organic matter co-deposited with the ore[81]. It has also been suggested that organic carbon in the reef zone is the result of hydrothermal organics migration through fractures extending from the deeper and lower porosity shales of the West Rand[82]. If the brines received any organics input from the Vaal Reef, this would also require contribution from the lower shale units of similar but lower overall organics composition[82]. Previous evidence, however, of high radiogenic activity for the Moab Khotsong system strongly suggests radionuclide contribution from the Vaal Reef[37,38]. As a result, a sample of the Vaal Reef was analyzed as a potential organics source for the brines throughout this study.

### Fracture fluid sampling
All fracture fluid and gas samples were collected during August 2018, August 2019, or January 2020 sampling trips from previously drilled boreholes in the mine; all sample analyses were performed on samples from the 2019 field expedition (the longest expedition at 6 weeks length, from which the most sample material was collected) unless otherwise specified. All boreholes (1200, 95, and 101-levels) were drilled originally using drilling fluid comprised of mine service water (low salinity water with input water from the Transvaal dolomite aquifer, which is circulated and cooled from the surface for use throughout the mine in general cleaning and drilling purposes)[83], plus a barium-based drilling grease (AMC Gorilla Grip), drilling lubricant (AMC Ezy-Core), and degreasing fluid (AMC Triple 4). The organic compositions of these drilling additives are included in this study (Supplementary Information Fig. S3 and Fig. S5) to evaluate potential contamination. Prior to sample collection from each site, fluid was flushed from the borehole for a minimum of 4 min. Collection of fluid at the 95-level occurred via an Inconel U-tube and packer device installed in 2018[84]. At the time of U-tube installation, fluorescein dye (~9 mM) was injected into the 95-level borehole to act as a chemical tracer of contamination from the installation; no fluorescein has been added to the borehole since the original installation in 2018. Following each sampling from the 95-level U-tube, the Inconel tubing was flushed with $N_2$ and sealed. Fracture fluid from the 1200 and 101-levels was collected with the use of a multi-port sterilized stainless steel manifold in lieu of a U-tube. The 101-level brine borehole was left exposed to air between drilling and sample collection in 2018 and subsequently capped with a steel cover until the 2019 sampling event in which the cap was replaced with a stainless-steel valve. Fluid at the 1200-level continuously flows out of an uncapped rubber hose at a rate of ~15 L/min. The mine service water was collected from a spigot on the 95-level. Fluid sampling events were coordinated with Moab Khotsong mine management and all materials and equipment were logged with the mine management upon entering and exiting the campus.

The Vaal Reef sample was collected as a section of rock from the main reef ore zone in the Moab Khotsong mine. A Precious Metals Special Permit was issued to and used by the University of the Witwatersrand to ship a 5-g section of the Vaal Reef to Princeton University. This 5-g portion of rock was ground in a sterile mortar and pestle, and extracted in 50 mL equal parts HPLC grade MetOH

(Methanol for HPLC Analysis, J.T. Baker) and $CH_2Cl_2$, (dichloromethane, Thermo Fisher Scientific) for 12 h. The solvent was fully evaporated in a fume hood under a gentle stream of $N_2$, and re-diluted in MQ $H_2O$ (Milli-Q purified $H_2O$, Millipore-Sigma).

Samples for quantification of low molecular weight organic acids were filtered on site (0.2 μm) into 15 mL Falcon tubes and frozen on surface (−20 °C). Fluid samples for all other organics analyses were filtered on site (0.2 μm) into 1 L amber vials and frozen on surface (−20 °C). All glassware was acid washed overnight (10% HCl), followed by combustion overnight at 450 °C[23,35]. Gas samples for volatile hydrocarbon quantification and $\delta^{13}C$, $\delta^2H$ analyses were collected in 2018 with the use of a gas-stripping device, and filled into 160 mL pre-evacuated serum vials. Volatiles measured via GC-MS were taken from headspace gas of the 1 L DOC amber vials for each sample.

## Measurements of dissolved organic carbon (DOC)/Dissolved Inorganic Carbon (DIC), low molecular weight organic acids, and $\delta^{13}C$ and $\Delta^{14}C$ of DOC/DIC

DOC and DIC collected in 2018 were measured from 0.2 μm filtered samples using an Aurora 1030 W TOC Analyzer (OI Analytical, College Station, Texas, USA) in the Onstott Lab at Princeton University. These measurements included acidification of DIC to $CO_2$ gas using a 0.5:8 mL ratio of 5% phosphoric acid solution to sample and reacted at 70 °C for 2 min. For lower salinity samples (1200-level and service water), acidification was followed by DOC oxidation to $CO_2$ using a 2:8 mL ratio of 10% sodium persulfate solution to sample and reacted at 98 °C for 3 min. For brine samples (95 and 101-levels), the oxidation protocol was optimized for potential $Cl^-$ interference by using an 8:2 mL ratio of 20% sodium persulfate solution to sample, reacted at 98 °C for 5 min. DIC and DOC were measured in mg C/L from the resulting $CO_2$ of the acidification and oxidation steps, respectively (detection limit 0.1 ppm with maximum reproducibility of 1% RSD). Very low DOC was found in measurement of a degassed MQ $H_2O$ blank + reagents signature (1.2 mg C/L), with no detectable DIC; this DOC signature was subtracted from all reported DOC and DIC sample values in this study. The $\delta^{13}C\text{-}CO_2$ compositions were measured for DOC using a Picarro G2101-i Isotopic $CO_2$ cavity ring down spectrometer (CRDS) with an operational range of 200–3500 ppm (Picarro Inc., Santa Clara, CA, USA). Isotopic measurements were made on sample $CO_2$ passed directly from the Aurora 1030 W to a Picarro Caddy Continuous Flow Interface (A2100) following quantification of DOC or DIC with a reproducibility of ±0.3‰ V-PDB for individual measurements. Values of $\delta^{13}C$ for DOC were normalized against a range of two powder glutamic acid DOC standards (USGS RSL, Reston, VA, USA; Standard 40 [−26.39‰] and Standard 41a [36.55 ‰]) prepared in either MQ $H_2O$ or in sterile, filtered (0.2 μm) 4 M $Cl^-$ brine. Normalization to the standards was applied using Eq. 3 in ref. 85. and values are slightly heavier compared to values in ref. 38 that were corrected against a gas-based standard. $\Delta^{14}C$ analyses of the DOC and the DIC were performed on samples collected in 2019 and stored in 500 mL bottles using the accelerator mass spectrometry (AMS) facility at the National Ocean Sciences accelerator mass spectrometer (NOSAMS) at Woods Hole Oceanographic Institution. $\delta^{13}C$ of DIC was additionally reported by NOSAMS via acidification and isotope ratio mass spectrometry. Stripping of the $CO_2$ via acidification involved passing of the fluid sample with an $N_2$ carrier gas through an extraction loop with two cold traps (one for water at −80 °C and one for $CO_2$ at −190 °C). 4 mL of an 85% phosphoric acid solution was introduced at the beginning of sample circulation, and stripping was allowed to occur for 10 minutes, resulting in 95% yield of sample $CO_2$[86]. This $CO_2$ was then analyzed using a VG Prism stable isotope ratio mass spectrometer with an uncertainty of ±0.2‰ V-PDB for individual measurements.

Low molecular weight organic acids were measured using a Dionex ICS-5000+ capillary HPLC system (Thermo Fisher Scientific, Waltham, MA, USA) in the Onstott Lab at Princeton University. Anions were

separated using a Dionex IonPac AS15 analytical column (2 × 250 mm inner diameter, I.D.), Dionex IonPac AG15 guard column (2 × 50 mm I.D.) and Dionex ADRS 600 suppressor (2 mm; current = 52 mA) with an eluent of potassium hydroxide at a flow rate of 0.35 mL/min using a gradient from 1 mM to 60 mM for 70 min. Samples for the Kloof Mine (younger, mesophilic brine) from the Witwatersrand Basin[35] and Kidd Creek[20] are included for inter- and outer- basin comparisons in the reporting of these results (Fig. 1). DOC/DIC quantities and their $\delta^{13}C$ signatures were measured on a Shimadzu TOC-VCSH carbon analyzer and Picarro G2101-I CRDS (±1% DOC/DIC and ±0.3‰ V-PDB $\delta^{13}C$) and an Aurora 1030 C high temperature catalytic conversion DOC analyzer coupled to a continuous flow IRMS (±20% DOC/DIC and ±2.5‰ V-PDB $\delta^{13}C$) for Kloof Mine[35] and Kidd Creek[1], respectively. These fluids (and others) are included throughout the study where complementary DOC analysis is available.

## UV-Vis and excitation emission matrices (EEMs) analyses

Optical properties of DOC were characterized in the Onstott (UV-Vis) and Myneni (EEMs) labs at Princeton University. Ultraviolet–visible (UV–Vis) absorbance was determined for 3.5 mL filtered DOC samples loaded into a 1-cm quartz cuvette and scanned from 200 to 700 nm. Corresponding EEMs fluorescence spectra were obtained for samples using a dual fluorometer. Spectra were collected with an emission range from 280 to 600 nm with a step size of 2 nm and an excitation range from 250 to 450 nm with a step size of 5 nm, and an integration time of 0.5 s. MQ $H_2O$ was used as the absorbance and fluorescence blank. Blank, inner filter, and Rayleigh and Raman scattering signal effects were removed from sample EEMs spectra following the PARFAC analysis data correction methods and staRdom package (v. 1.1.14) in RStudio (v. 4.0.0)[87,88]. Samples of the 95-level were analyzed for 2018, 2019, and 2020 collection years to observe the changes in fluorescein overtime.

## [D]-, [L]- Aspartic acid hydrolysis and quantification via HPLC

Protein hydrolysis was performed on fracture fluid samples for determination of D- and L-aspartic acid[50,51]. 42.5 mL of frozen fluid each was used from the 95-level and 1200-level, while 5 mL was used for hydrolysis of the 101-level brine due to less sample collected in the field. The samples underwent two rounds of centrifugation for 20 min at 10,000 × g and 4 °C, with supernatant discarded between rounds; 300 μL of the removed supernatant was saved for hydrolysis of each sample. The remaining "pellet" was resuspended in 100–200 μL of 6 M HCl and gas-tight glass vials were flushed with $N_2$ prior to being incubated for 16 h at 100 °C. Vials were then opened and dried at 50 °C. Remaining hydrolysate was resuspended in 500 μL of MQ $H_2O$. For the 95 and 101-level brines, 10 μL of 10 M NaOH was added to prevent inhibition by high iron concentrations. Resuspended samples were vortexed, then centrifuged for 5 min at 17,000 × g and 4 °C. The supernatant was removed and used for subsequent HPLC analysis. Samples (200 μL) were first derivatized with a stock solution (50 μL) of o-phthaldialdehyde (OPA) and N-acetyl-L-cysteine (NAC). This mixture was vortexed and allowed to stand for 5 min before addition of 750 μL 50 mM sodium acetate (0.2 μm-filtered, pH 5.4). D- and L-enantiomers of aspartic acid were analyzed in the final solution using a HPLC system (PU-1580, JASCO, Japan) and attached fluorescence detector (FP -1520). Enantiomers were separated on a Waters C18 column (Nova Pak@ c18, 300 by 3.9. mm with 5 mm particle size) along with a mobile phase of 100% methanol (HPLC grade, Sigma) and 50 mM sodium acetate buffer at pH 5.4[51]. The gradient conditions for the mobile phase (sodium acetate:methanol) included the following introduced at a steady flow rate of 0.6 mL/min over a 33 min run: 0 min (98%:2%), 5.5 min (90%:10%), 16 min (80%:20%), 18 min (5%:95%), 23 min (5%:95%), 25 min (98%:2%), 33 min (98%:2%)[51]. The resulting D/L ratios were corrected to account for the 0.6 to 2.0% racemization caused by the $N_2$ flush hydrolysis method, as presented in ref. 50. Concentrations were

normalized to reflect differences in the volume of original sample hydrolyzed. Aspartic acid concentrations were used to estimate cellular densities for each fluid sample:

$$C = \frac{A}{0.0537 \cdot 0.524 \cdot (3 \cdot 10^{-13})} \tag{1}$$

Where C represents cellular density in cells/mL, A represents aspartic acid concentration in g/mL, 5.37% represents the average relative fraction of aspartic acid in aquatic bacterial cells from ref. [89], 52.4% is assumed as the cellular percent protein content (by mass)[90], and $3 \cdot 10^{-13}$ g was estimated as the cellular weight[91]. This calculation also assumes all aspartic acid is associated with intact microbial cells. The percent of amino acids comprising each DOC pool was estimated from amino acid percentages of moderate halophile *Chromohalobacter*[90]. This genus of halophile was selected as it has been identified in several other crustal subsurface brine systems[92,93].

### Quantification and isotopic analysis of volatile organics

Volatile organics were determined at ExxonMobil's Clinton campus with a qualitative gas chromatograph-mass spectrometer (GC-MS) run on the headspace gas from 1 L DOC samples. Identified volatile alkanes (methane, ethane, etc.) were measured from 160 mL serum gas vial samples using a Varian 3400 gas chromatograph (GC) with a flame ionized detector (FID). The gases were separated using a He carrier and J&W Scientific GS-Q column (30 m × 0.32 mm ID)[24]. An initial temperature of 60 °C was held for 2.5 minutes, followed by a 5 °C/min increase to a final temperature of 120 °C[24].

Samples for hydrocarbon gas analysis were collected in 2018 for the 95 and 101-levels, and their isotopic compositions determined in the Stable Isotope Laboratory at the University of Toronto[1,24,29]. $\delta^{13}C$ (with respect to V-PDB) was measured using a gas chromatograph combustion isotope ratio mass spectrometry (GC-C-IRMS)[1,24,29]. This analysis used a Poraplot Q™ column (25 m × 0.32 mm ID) with a Finnigan MAT 252 mass spectrometer coupled to a Varian 3400 capillary GC. Hydrocarbons were separated with an initial temperature of 40 °C held for 1 min, followed by a 5 °C/min increase (held for 5 min) to a final temperature of 190 °C. The error for measurements was ±0.5‰.

$\delta^2H$ (with respect to V-SMOW) was measured for hydrocarbons using a continuous flow compound specific hydrogen isotope mass spectrometer[1,24,29]. Hydrocarbon gases were separated on a Poraplot Q™ column (25 m x 0.32 mm ID) and analyzed with a 6890 gas chromatograph (GC) coupled to a micropyrolysis furnace (1465 °C) and connected in line with a Finnigan MAT Delta⁺-XL isotope ratio mass spectrometer. The He carrier was maintained at 2 mL/min, with an initial temperature of 35 °C held for 3 min, followed by a 15 °C/min increase to a final temperature of 180 °C. The error for measurements was ±5‰.

### (Negative) Electrospray ionization Fourier-transform ion cyclotron resonance mass spectrometry (−ESI 21 tesla FT-ICR MS)

Samples used for solid phase extraction (SPE) were taken from the same 1 L amber vials of 0.2 μm filtered sample fluid that were collected and used for DOC/DIC quantification on the Aurora TOC analyzer. An aliquot of the Vaal Reef was taken from the prior aqueous extract for this sample. SPE was performed by passing sample fluids through Agilent Bond Elut PPL cartridges (200 mg, 3 mL) under vacuum. Cartridges were first prepped by running through 15 mL MQ H₂O (acidified to pH 2 with HCl) followed by 15 mL HPLC grade MetOH. The following volumes were run for each fluid samples on prepped cartridges, based on DOC quantity and available sample remaining: 95-level (30 mL), 101-level (15 mL), 1200-level (100 mL), Vaal Reef (30 mL), service water and drilling additives (30 mL each), and a MQ H₂O blank (30 mL). Samples were followed by an additional 15 mL rinse with acidified MQ H₂O

(this volume was doubled for brine samples). Cartridges were left to dry overnight in a fume hood. All samples were gravity-eluted with 9 mL of HPLC grade MetOH the next day. Vials were left open in the hood under a gentle stream of N₂ until the fluid evaporated to a final volume of 4 mL. The extraction efficiency on average for this procedure was 65%.

Final SPE extracts were sent to the National High Magnetic Field Lab (Tallahassee, FL) and analyzed via negative electrospray ionization with a custom-built 21-tesla Fourier-transform ion cyclotron resonance mass spectrometer[94]. Sample solution was infused via a microelectrospray source (50 μm i.d. fused silica emitter) at 500 nL/min. Typical conditions for negative ion formation were: emitter voltage: −3.0 kV; S-lens RF level: 45%; and heated metal capillary temperature: 350 °C. Ions were analyzed with a custom-built hybrid dual linear RF ion trap FT-ICR mass spectrometer equipped with a 21 T superconducting solenoid magnet. Ions were initially accumulated in an external multipole ion guide (1-5 ms) and transmitted to the ICR analyzer cell in a m/z-dependent axial ion ejection process that limits time-of-flight distortion[95]. The target number of charges was set to $1 \cdot 10^6$, which was assisted by automatic gain control[96]. Ions were excited to m/z-dependent radii to maximize the dynamic range and number of observed mass spectral peaks[97]. The dynamically harmonized ICR cell in the 21 tesla FT-ICR was operated with 6 V trapping potential[98]. Time-domain transients of 3.1 s were acquired with a Predator data station[99] that handled ion excitation and detection only, initiated by a TTL trigger from a commercial Thermo data station, with 100 time-domain acquisitions averaged for all experiments. Mass spectra were phase-corrected and internally calibrated with 10 highly abundant homologous series that span the entire molecular weight distribution based on the "walking" calibration method[100]. Molecular formulas with an error <0.3 ppm were assigned, and chemical classes with a combined relative abundance of ≥0.15% of the total were considered for the molecular composition reported in the work herein. Molecular formula, DBE, m/z assignments and Van Krevelen diagrams were made using PetroOrg software developed by the National High Magnetic Field Lab (https://nationalmaglab.org/user-facilities/icr/icr-software)[101].

### Reporting summary

Further information on research design is available in the Nature Portfolio Reporting Summary linked to this article.

## Data availability

The supporting UvVis, EEMs, hydrocarbon $\delta^2H$ and $\delta^{13}C$, and FT-ICR MS data generated in this study, alongside investigation of $^{14}C$ contribution pathways, are provided in Supplementary Information. FT-ICR MS spectra for samples presented in this study are included in Supplementary Data.

## Code availability

This study only utilized published code from previous studies. Specifically, it utilized the following code/software: PARAFAC analysis using staRdom package (v.1.1.1) (EEMS analysis)[87], RStudio (v.4.0.0)[88], and PetroOrg Software (v.18.0.6)[101].

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

## Acknowledgements

This study was financially supported by the National Science Foundation grants EAR2026858 (to TLK) and EAR2026853 (to TCO). Additional financial support was provided by a National Science Foundation Graduate Research Fellowship, a NASA Astrobiology Early Career Award, and a Walbridge Fund Graduate Research Award (from Princeton University's High Meadows Environmental Institute) to DMN. A portion of this work was performed at the National High Magnetic Field Laboratory, which is supported by the National Science Foundation Cooperative Agreement No. DMR-1644779 and the State of Florida. Further support for hydrocarbon analysis was awarded through Natural Sciences and Engineering Research Council of Canada Discovery and Accelerator grants to BSL with additional funding by CIFAR to Earth 4D Director and Fellow BSL. The grants from ICDP, JSPS Core to Core Program, and MEXT Japan to H. Ogasawara of Ritsumeikan University, Japan and the DSeis international team supported the cost of drilling the borehole, downhole logging, and core curation at the Moab Khotsong Mine. We thank B. Liebenberg and the staff of Lesedi Drilling & Mining (Pty) Ltd and Van Heerden Esterhuizen and Brenda Freese of Moab Khotsong Mine and the management of Harmony Gold Mining Company Ltd. for their logistical support. We thank Z. Garvin of Princeton University and Dr. J.-G. Vermeulen of the University of the Free State for deploying the U-tube sampler. Finally, we thank Prof. E. van Heerden and C. van Vuuren of iWater alongside Jameel Alom of the University of the Free State for help with the collection and shipment of samples and TIA (Technology and Innovation Agency, South Africa, SABDI 16/1070) for supporting Dr Julio Castillo with funding for samples collection.

## Author contributions

D.M.N., C.C.W., T.L.K., and T.C.O. contributed to the design of this study. D.M.N., T.L.K., O.W., J.C., E.C., and T.C.O. contributed to sample collection. B.M.F. contributed the U-tube sampling system for 95-level brine sample collection. D.M.N., C.C.W., T.L.K., M.C., and C.W. contributed to the data prep, acquisition, and interpretation of negative ion ESI 21 tesla FT-ICR MS data at the National High Magnetic Field Lab. B.S.L. and O.W. contributed to the sample collection and hydrocarbon quantification and isotopic data measured at the University of Toronto. D.M.N., C.C.W., T.L.K., B.S.L., O.W., J.C., S.P., and T.C.O. contributed to the interpretation of other organics analyses presented here. D.M.N. wrote and organized the manuscript with contribution from all coauthors.

## Competing interests

The authors declare no competing interests.
