## [Peer Review File · Nature Communications]

Radiolytically reworked Archean organic matter in a habitable deep ancient high-temperature brineReviewer #1 (Remarks to the Author):

The manuscript titled 'Radiolytically reworked Archean organic matter enables life in physiologically challenging deep ancient and thermal brine' by Nisson and colleagues presents comprehensive analyses of organic compounds from ancient subsurface brines that shows how such environments can host a diverse range of organic transformations driven by radiolysis that can supply energy to sustain biotic processes. This is an exciting finding that has implications for understanding subsurface habitability on Earth and elsewhere in the solar system. The authors integrate an impressive range of analytical techniques and the manuscript is at its strongest when describing the likely origins and fate of dissolved organic compounds. Some aspects regarding microbial habitability and survival are more speculative and could benefit with further clarification and/or caveats.

I have some recommendations I feel should be addressed before publication.

(1) There were some instances of field-specific jargon or assumed reader knowledge. I felt the manuscript would benefit from edits in several places to facilitate broader understanding by the interdisciplinary readership of Nat Comms. Specific instances:

(1a) The use of the word 'thermal' in the title and elsewhere as an adjective to refer to a high-temperature brine (Title, Lines 47, 508) might be misleading to some readers, as it can also imply a fluid which has a (hydro)thermal origin, but which is no longer at high temperatures. I suggest taking describing the brine as 'high-temperature' (or similar wording).

(1b) Despite radiation being mentioned in the title and abstract, and forming a vital part of the narrative, the introduction does not discuss the radiation environment in the Moab Khotsong brines or the sources of the radiation. These aspects are left to the reader to infer until much later in the manuscript (first explicit mention of uranium decay is on line 420). The introduction should include a description of the radiation and its likely sources.

(1c) Most sections of the results assume prior knowledge of the techniques and/or the specific approaches/measurement strategies used in the study. The necessary information is in the Methods, but for ease of understanding by the reader, each aspect of the Results should begin with a short preamble that mentions what analysis was carried out and why. This should not duplicate the detailed Methods, but provide a brief primer on each approach and what information this provides specifically for this study. Lines 204-207 is a partial example of this approach for ESI 21 tesla FT-ICR MS.

Specific locations needed:

Line 147: Some brief description of sampling locations and geological settings of the brines should be included before the authors move into shorthand designations.

Line 175: Some preamble is required to describe how and why EEMs were constructed

Line 204: Define acronyms on first use. Also include a brief mention of what this technique facilitates (e.g., produces high-res mass spectra for compound identification).

Line 323: Preamble is essential here. Currently it's very unclear what is being described here- is this something from the literature? Model results?

Line 340: Briefly summarise how microbial survival was calculated and include appropriate caveats (relates to below point).

(2) Currently, conclusions about the microbial communities, their metabolic function and their long-term survival are speculative as the authors have not characterised the microbial groups present. While the free energy calculations are helpful, and add to the

manuscript, community analyses (e.g., nucleic acid-based or cultivation-based) would be necessary to support some of the statements made in the manuscript. The authors should be clear about where the limitations of their extrapolations are and temper conclusions accordingly. Specific locations:

(2a) Line 372: "The organic matter in the brines is able to support a low biomass microbial community" – The authors have not directly demonstrated that the microbial community is sustained by the organic compounds. The authors should summarise the main line(s) of evidence that an active microbial community is present to support this statement, e.g., 'compounds of microbial origin were detected, certain metabolisms are thermodynamically feasible, suggesting that OM in the brines supports microbial community'

(2b) Lines 480 – 48: Thermodynamic calculations are just predictions – without community data that directly demonstrates presence/absence of certain metabolising groups, it's not possible to draw firm conclusions about habitability. I would like to see the authors be clear that these are predictions (interesting ones) but more data is required to test them.

(2c) Lines 492 – 495: These survival times are highly speculative. The authors are not able to determine how radiation-resistant the communities in the brines are without community analyses. I would prefer that the authors are very clear that these are illustrative examples of how specific organisms would respond if exposed to these doses, and not an indication of the longevity of the actual microbes in situ. The authors also need to describe how these calculations of survival were performed (in the Methods, see below).

(2d) Lines 507-508: "...allow the Moab Khotsong brine system to host a novel community structure" – the authors cannot comment on community structure with their current data. They can instead make predictions about what metabolic groups should be present based on thermodynamic calculations. This statement should be modified to reflect the uncertainty.

(2e) Line 510: "presence of a low biomass, micro-aerophilic community" this has not been directly observed- it is an inference from thermodynamic predictions. This statement should be modified to reflect the uncertainty.

(3) Lines 746 – 771: It's not clear how microbial survival times were calculated. These should be described here.

Other minor comments:

Line 27: add a qualifier to mention the source of the radiation

Line 29-30: Define acronyms on first use, or use alternative term for abstract (e.g., 'spectroscopic and mass spectrometric approaches suggest...')

Line 53 – Greater diversity than what? Please clarify

Lines 113 – 114: The authors might consider using the term 'stress' instead of thermodynamic constraints- not all stresses imposed by salinity, temperature, etc. will relate to thermodynamic constraints.

Line 116: It's not clear what is being compared here – higher biomass than what? Please clarify

Line 126: In what sense are these settings both 'isolated' and 'global'? Please clarify and add a citation if necessary.

Figure 1: $\delta^{13}\text{C}$ arguments in the Discussion would benefit from plotting on here

expected ranges from various biotic/abiotic processes drawn from the literature.

Line 194: Should be 'spectrum' (singular)

All figures/tables: I suggest moving these to after their first mention in the text

Lines 483 – 486: It's not clear what setting you are describing here. Please clarify where you expect to find cryogenic brines in the subsurface.

Lines 514 – 515: It's not clear what setting on Europa and Enceladus the authors are describing here. The temperatures at 1m depth on Europa and Enceladus will be $\leq 100\text{k}$. There will be no liquid water and even volatiles like methane will be frozen.

Lines 520: I suggest the authors consider implications for Ceres, where subsurface brines have been inferred.

Line 725: Seems very high. Should this be mmol?

Reviewer #2 (Remarks to the Author):

The Nisson et al. manuscript on the ancient organic matter pool feeding the deep biosphere in brines of South African mine informs us about the potential habitability and possible microbial metabolisms on this extreme environment. The results show the importance of radiolytic reworking of ancient organic carbon pool of host rocks to microbial communities, in addition to already established radiolytically formed electron acceptors such as oxygen and nitrogen. Interestingly, the microbial metabolism this environment seems to best support is microaerophilic heterotrophy. Similar ideas have been previously suggested at least for the Fennoscandian deep biosphere. Thus, the study gives further confirmation about the importance of heterotrophy in deep terrestrial subsurface habitats. The abiotic water-rock interactions are the main source of organic matter that the low cell number -microbial communities are able to utilize, and according to energy flux calculations, again, heterotrophic metabolism combined with aerobic or nitrate reduction provides enough energy for the microbes to survive long time periods in the hypersaline and hot deep subsurface environment. Alongside, the authors also modelled the capacity of well-known radiotolerant and -resistant bacterial species to survive in these conditions, leading to estimate of survival up to 750000 years by the most radioresistant microbes.

I found the manuscript well-prepared, clear and concise, and results supported the conclusions. Some methods (EEMs and negative ion ESI 21 tesla FT-ICR MS) are somewhat unfamiliar to me, hampering the assessment of their relevancy, but the results are clearly explained and credible. I have only minor details that the authors might want to take into consideration, but I leave it up to authors to decide if they found these useful:

-adding a bit more detail on the discussion about heterotrophic metabolism in the deep biosphere. There are a few studies where this has been touched and this could be elaborated in the manuscript. In addition, astrobiological discussion on habitability has concentrated on serpentinization and autotrophy, but this manuscript shows another potential energy and carbon source for extraterrestrial microbes.

-adding some comparison about the time frame the microbial community could have survived in the Moab Khotsong deep subsurface in contrast to other old fluids in Canada, SA and Fennoscandia.

- some more details on the lithology and fracturing of the rock would be great. Was Vaal Reef sample analyzed because it might be a potential source of the DOC pool in Moab? Is there any knowledge on the composition of shales surrounding the 95 and 101 sites in the likely more porous sandstone units? Could those also host a pool of organic matter?

Best,
Lotta Purkamo, Geological Survey of Finland

Reviewer #3 (Remarks to the Author):

Summary

This is a very thorough and interesting study of carbon cycling and life in a deep subsurface ecosystem that has been isolated from the surface world for an extraordinarily long interval of time (>1 Ga). It provides novel insight into one of the most enigmatic ecosystems on Earth. Many of its results are noteworthy and will be significant to the field (see conclusions 1-3 below). With appropriate major revision, it will be very appropriate for publication in Nature Communications.

The primary conclusions are as follows:

1. EEMs, negative ion ESI 21 tesla FT-ICR MS, and amino acid analyses suggest the brines likely contain a large geologically "old" biotic carbon pool, derived from radiolytically oxidized kerogen-rich shales or reefs, that overshadow current microbial contribution.
2. The GC and isotopic analysis of volatiles reveal predominately n-alkanes (<C6) of methane and ethane, and C1-C3 hydrocarbon $\delta^2\text{H}$ and $\delta^{13}\text{C}$ signatures consistent with an abiotic origin.
3. These findings suggest that over the long subsurface isolation of the brine, water-rock processes control redox and C cycling, lending support for a low biomass, "slow biosphere" of micro-aerophilic and heterotrophic microbes supported by radiolytic oxidants.

These primary conclusions are generally well-supported. They constitute a substantial advance in understanding of deep subsurface life.

The evidence for the geologically "old" biotic carbon pool and its origin appears solid. The ^{14}C signature of this carbon was initially surprising to me, but the reporting summary makes a reasonable case that the ^{14}C was created primarily, perhaps entirely, in situ (as suggested at the top of page 22).

The evidence for an abiotic origin of the n-alkanes (<C6) of methane and ethane, and C1-C3 hydrocarbon $\delta^2\text{H}$ and $\delta^{13}\text{C}$ signatures also appears solid.

The evidence for low biomass requires more nuanced discussion, but the very low biomass estimates are likelier to be too high than too low. See below for details.

The inference of micro-aerophilic and heterotrophic microbes supported by radiolytic oxidants is reasonably well supported by the energies of reactions.

Primary concerns

My primary concerns are associated with the calculations and assumptions focused on microbial metabolism, maintenance, and cell densities. The cell density calculations require clearer and more thorough discussion in a revised manuscript. The "free energy flux" calculations and the cell maintenance calculations pose more intractable problems.

These calculations are not central to the primary conclusions (1-3 above). As they're presently calculated and presented, they significantly weaken the manuscript.

1. Cell density estimates are derived from aspartic acid concentrations (Table 1 and page 16). This derivation appears to implicitly assume that the aspartic acid is solely associated with intact cells, rather than with dead biomass or in dissolved form. This assumption should be explicitly acknowledged. Primary contribution from dead biomass would be consistent with the page-24 mention that "Cell density estimates based on aspartic acid concentration were 1-2 orders of magnitude greater than original Syto-9 or Raman-based cell counts for each sample (from reference 39). The abstract appears to implicitly acknowledge this point by giving a cell density range that is 2 to 5 orders of magnitude lower than the estimated densities in Table 1 (for Level 95 filtered and Level 1200, respectively). The difference between the Table 1/Results aspartic-acid-based estimates and the abstract estimates should be clearly explained by discussing this topic more thoroughly in the manuscript.

2. It's surprising to me that racemization age estimates were not calculated for the aspartic acid enantiomeric ratios. Why was this not done? Would it strengthen or weaken the arguments of the manuscript?

3. The "free energy flux" calculations appear very problematic. Based on the page-38 description, free energy flux is calculated from assumptions of cell radius, diffusivity to the cell center, and constant concentrations of the reactants, with no consideration of reactant flux rates to the ecosystem or product flux rates from the ecosystem. This calculation appears to assume that reactants for all the cells enter the environment and products from all the cells exit the environment as fast as the "limiting substrate" can diffuse the distance of the assumed cell radius. This is a huge assumption. It seems unlikely to be true since timescale of diffusion is directly related to the square of the distance, a cell radius is very small, and the ultimate sources and sinks of the reactants and products are probably far from cells. By assuming a fixed "substrate" concentration and a fixed in situ free energy, the calculation also assumes that reactant and product concentrations are constant, although the reactions remove the reactants and produce the products. These assumptions can only be true if reactants and products are replaced and removed as rapidly as they are respectively used and created. Without quantitative consideration of the rates at which reactants enter and products leave the larger ecosystem, the cell-radius "free energy flux" calculations seem unlikely to capture the true flux of free energy to this >1-Ga ecosystem or the cells that inhabit it. I don't see a way to resolve this problem without calculating energy fluxes in a way that explicitly accounts for how fast reactants enter the 95-level system as a whole.

Why is 80% of the calculated free energy assumed to be available for metabolism? This is a high value.

4. The maintenance energy calculations also appear very problematic. Maintenance energy is defined in very different ways by different authors and for different equations. Most of the equations used, including the one used in this manuscript, are not demonstrably relevant for calculating microbial growth or survival in natural ecosystems with very slow life. For a fairly thorough review of maintenance energy calculations and their associated problems (including for the equation used by this manuscript), see, e.g., P. van Bodegum, 2007 (Microbial Maintenance: A Critical Review on Its Quantification, Microbial Ecology, Vol. 53, No. 4, 513-523).

The authors could most simply resolve concerns 3 and 4 (above) by deleting these topics (free energy fluxes and maintenance energy) from the manuscript. They are not central to the strong primary conclusions of the manuscript and they significantly weaken the manuscript in their present state. If the authors wish to retain these topics, they should either recalculate them using approaches that meet the fundamental concerns outlined above or explicitly describe the problems inherent in the present calculations and explicitly treat the calculated values as model predictions, rather than

as definitive descriptions of free energy fluxes and maintenance energy in this extraordinary ecosystem.

Secondary concerns

Why include free energy calculations for 57 total reactions when only 10 are discussed as favorable?

Minor points:

Acronyms, such as EEMs and FT-ICR MS should not be used in the abstract or the manuscript without being defined when they're first used.

REVIEWER COMMENTS

Response to Reviewer Comments is presented in blue text. Edited lines refer to the manuscript version with tracked changes. Most citations are included in the references section of the main text; a doi has been provided for those that do not appear in the main text.

Reviewer #1 (Remarks to the Author):

The manuscript titled 'Radiolytically reworked Archean organic matter enables life in physiologically challenging deep ancient and thermal brine' by Nisson and colleagues presents comprehensive analyses of organic compounds from ancient subsurface brines that shows how such environments can host a diverse range of organic transformations driven by radiolysis that can supply energy to sustain biotic processes. This is an exciting finding that has implications for understanding subsurface habitability on Earth and elsewhere in the solar system. The authors integrate an impressive range of analytical techniques and the manuscript is at its strongest when describing the likely origins and fate of dissolved organic compounds. Some aspects regarding microbial habitability and survival are more speculative and could benefit with further clarification and/or caveats.

I have some recommendations I feel should be addressed before publication.

(1) There were some instances of field-specific jargon or assumed reader knowledge. I felt the manuscript would benefit from edits in several places to facilitate broader understanding by the interdisciplinary readership of Nat Comms. Specific instances:

(1a) The use of the word 'thermal' in the title and elsewhere as an adjective to refer to a high-temperature brine (Title, Lines 47, 508) might be misleading to some readers, as it can also imply a fluid which has a (hydro)thermal origin, but which is no longer at high temperatures. I suggest taking describing the brine as 'high-temperature' (or similar wording).

The term 'thermal' used in the title and throughout the manuscript to describe fracture fluid temperature has been changed to 'high temperature' or similar phrasing where applicable to avoid confusion with the terminology.

Lines edited: 2 & 32

(1b) Despite radiation being mentioned in the title and abstract, and forming a vital part of the narrative, the introduction does not discuss the radiation environment in the Moab Khotsong brines or the sources of the radiation. These aspects are left to the reader to infer until much later in the manuscript (first explicit mention of uranium decay is on line 420). The introduction should include a description of the radiation and its likely sources.

We agree with the reviewer's suggestion that the sources of radiation should receive additional description in the introduction. As a result, several lines have been added to the introduction regarding radionuclide sources, where descriptions of abiotic stress in the Moab Khotsong brines first appear. This added description includes mention of both decay from local radionuclide species (≤ 1.1 ppm U, ≤ 3.2 ppm Th, $\leq 0.50\%$ K) based on core measured values

from Nisson et al., 2023, as well as additional contribution from the U-enriched Vaal Reef zone (which is mentioned throughout the manuscript also for its kerogen enrichment). Specifically, we include mention of the minimum estimated contribution from the reef (11-20 ppm based on noble gas- and $^{36}\text{Cl}/\text{Cl}$ -data from Warr et al., 2022, and Nisson et al., 2023, respectively) with potential contribution reaching up to 100 ppm based on the maximum $^{36}\text{Cl}/\text{Cl}$ basin estimate from Purtschert et al., 2015, and discussed in Nisson et al., 2023.

Edited Lines: 263-269

(1c) Most sections of the results assume prior knowledge of the techniques and/or the specific approaches/measurement strategies used in the study. The necessary information is in the Methods, but for ease of understanding by the reader, each aspect of the Results should begin with a short preamble that mentions what analysis was carried out and why. This should not duplicate the detailed Methods, but provide a brief primer on each approach and what information this provides specifically for this study. Lines 204-207 is a partial example of this approach for ESI 21 tesla FT-ICR MS.

We agree with the reviewer that several sections within the results require a clarifying preamble. In response, we have gone through this section of the manuscript and added a few sentences of general introduction wherever the results of a new technique are presented. This did not include the FT-ICR MS or radiolytic survival model sections, as these already contained a general introduction where they are presented in the results.

Edited lines: 382-385, 427-456, 598-602, 632-634, 701-715, 745-748

Specific locations needed:

Line 147: Some brief description of sampling locations and geological settings of the brines should be included before the authors move into shorthand designations.

A few sentences introducing the fluid systems considered, and their naming designations (e.g. 95-, 101-, and 1200-levels) have been added to the beginning of the results section. We also include brief mention of the surrounding geology and stratigraphic section for each of the sampling locations.

Edited lines: 374-380

Line 175: Some preamble is required to describe how and why EEMs were constructed.

We appreciate and agree with the reviewer's suggestion for clarification as to why EEMs analysis were utilized for this manuscript. A few sentences were added to the beginning of this section to clarify how EEMs spectra are obtained (dual scanning fluorometer, collection of fluorescence across a range of excitation and emission wavelengths), and how the peaks identified in visualization of these spectra can be compared to literature-based values of fluorescent DOC species. We reserved thorough discussion of converting the raw data into an EEM spectrum graphic for later in the methods section.

Edited Lines: 427-456

Line 204: Define acronyms on first use. Also include a brief mention of what this technique facilitates (e.g., produces high-res mass spectra for compound identification).

All abbreviations presented throughout the abstract and main text are now fully defined on their first use. If abbreviations appear in both the abstract and main text, they are now accompanied by a full definition of the term on their first appearance in both (abstract and main text). The first appearance of technique-based abbreviations in the main text now additionally include a brief description of the technique and what we are trying to accomplish by employing the technique.

Edited Lines: 35-37, 338, 598, 631, 702

Line 323: Preamble is essential here. Currently it's very unclear what is being described here- is this something from the literature? Model results?

We agree with the reviewer that this section requires a clarifying preamble. A few sentences have been added to the beginning of this paragraph explaining that the free energy flux parameters are a model-based calculation, drawing from previous models presented in the literature (Onstott et al. 2006, 2014; Lau et al., 2014; Magnabosco et al., 2016). The free energy flux values presented in this manuscript are unique to the 95-level brine at Moab Khotsong, as the fluid composition used to obtain in-situ free energy values for metabolic reactions requires inorganic and organic species compositions from the actual 95-level brine (many of which were measured and presented in Nisson et al., 2023, with inclusion of some organics presented in this manuscript). Additionally, the free energy flux equation incorporates the concentration of the limiting reactant for each metabolic strategy considered, conditions which are again unique to the composition of the 95-level brine. Calculations for maintenance energy are mentioned here as being based on literature-derived values (Tijhuis et al., 1993 and Oren et al., 1999). Specific description of calculations remain outlined the methods section.

Edited Lines: 701-715

Line 340: Briefly summarise how microbial survival was calculated and include appropriate caveats (relates to below point).

To address this point by the reviewer here and later (2c, Line 746) this section of the results has been edited in an attempt to more clearly convey how microbial 'survival times' were considered under the dosage/particle scenarios unique to Moab Khotsong. Each bacterial population was determined inactive, or unable to survive, at the accumulated dosage resulting in a 10^{-6} population reduction (previously determined in accumulated dosage experiments for *E. coli* (1,000 Gy) (Battista et al., 1997), *B. subtilis* (8,400 Gy) (Bhaumstak-Khan, 2002), and *D. radiodurans* (15,000 Gy) (Battista et al., 1997) cultures). The actual accumulated dosage at which each single-species population achieves a 10^{-6} survival fraction is taken from these previous laboratory-based studies in the literature. We simply apply these accumulated-dosage values to the unique dosage/particle scenarios of our field site (Moab Khotsong).

Edited Lines: 743-801

(2) Currently, conclusions about the microbial communities, their metabolic function and their long-term survival are speculative as the authors have not characterised the microbial groups present. While the free energy calculations are helpful, and add to the manuscript, community analyses (e.g., nucleic acid-based or cultivation-based) would be necessary to support some of

the statements made in the manuscript. The authors should be clear about where the limitations of their extrapolations are and temper conclusions accordingly. Specific locations: We appreciate the reviewer's suggestion to clarify and tone down our statements regarding microbial community composition for this site and have edited the manuscript accordingly. Please refer to our response on reviewer comments 2b, 2e for specific changes made in response to these statements throughout the manuscript.

(2a) Line 372: "The organic matter in the brines is able to support a low biomass microbial community" – The authors have not directly demonstrated that the microbial community is sustained by the organic compounds. The authors should summarise the main line(s) of evidence that an active microbial community is present to support this statement, e.g.,; 'compounds of microbial origin were detected, certain metabolisms are thermodynamically feasible, suggesting that OM in the brines supports microbial community'

We agree with the reviewer that the statements regarding Moab Khotsong microbial community composition need to reflect less certainty. We have altered this section of the discussion per the suggestions of the reviewer to state that the data presented in this study *may* suggest the brines are capable of supporting a low biomass community of heterotrophic microbes.

Edited Lines: 851

(2b) Lines 480 – 48: Thermodynamic calculations are just predictions – without community data that directly demonstrates presence/absence of certain metabolising groups, it's not possible to draw firm conclusions about habitability. I would like to see the authors be clear that these are predictions (interesting ones) but more data is required to test them.

We appreciate the reviewer's suggestion to clarify and tone down our statements regarding microbial community composition for this site. We certainly agree that thermodynamic predictions like these, only provide one estimate of potentially supported metabolic strategies under the unique chemistry of the Moab Khotsong brine system. As a result, we have added in a few sentences to the section in the discussion where free energy flux model results are presented. Here, we clarify that these are merely energetic predictions, and we would need future sequence-based annotation data to make more definitive statements regarding the composition of an *in situ* microbial community. We also believe the preamble added to the sections on free energy flux in the Results and Methods better introduce this technique as a predictive model.

Edited Lines: 701-715, 1238-1246, 1792-1818

(2c) Lines 492 – 495: These survival times are highly speculative. The authors are not able to determine how radiation-resistant the communities in the brines are without community analyses. I would prefer that the authors are very clear that these are illustrative examples of how specific organisms would respond if exposed to these doses, and not an indication of the longevity of the actual microbes *in situ*. The authors also need to describe how these calculations of survival were performed (in the Methods, see below).

The reviewer makes a good point here. To reflect this, the manuscript has been revised and now explicitly states that these are illustrative examples of how specific organisms would

respond if exposed to these doses on Lines 452, 985, and in the caption to Figure 7. Likewise, we also have added considerable additional detail, clarification, and context to the calculations in main text (Lines 450-460) and the Methods (lines 983-1004) as also requested by the reviewer. Lastly, we have revised the sentence on line 645-648 to reflect the uncertainty of applying these calculations to in situ microbiology and habitability.

Edited Lines: 450-460, 645-648, 983-1004

(2d) Lines 507-508: "...allow the Moab Khotsong brine system to host a novel community structure" - the authors cannot comment on community structure with their current data. They can instead make predictions about what metabolic groups should be present based on thermodynamic calculations. This statement should be modified to reflect the uncertainty. We agree with the reviewer that the confidence on statements regarding Moab Khotsong microbial community composition needs to reflect greater uncertainty and have edited the manuscript per their previous comment (2c). In addition to this, definitive statements such as "Moab Khotsong hosts a novel community" have been altered to say "The large contributions by radiolysis ... suggest the potential for the Moab Khotsong brine system to host a microbial community in which anaerobic/aerobic heterotrophs could be energetically supported". We additionally acknowledge in proceeding sentences that definitive statements on microbial composition cannot be made without sequence-based annotation data.

Edited Lines: 1317-1386

(2e) Line 510: "presence of a low biomass, micro-aerophilic community" this has not been directly observed- it is an inference from thermodynamic predictions. This statement should be modified to reflect the uncertainty.

Again, we agree with this point raised by the reviewer. Accordingly, this sentence has been re-written to state "The presence of a low biomass, community at Moab Khotsong that may be composed of thermodynamically supported micro-aerophilic heterotrophs,..." so that appropriate uncertainty is introduced. Likewise, additional clarification is now provided on Lines 621-630 and in the Methods (936-939) to specifically state that direct investigation into community composition and metabolic pathways being utilized here presently remains unresolved to reflect this point by the reviewer. That said, it can be stated still that the brines host a low-biomass microbial community, as these are supported by direct cell counts from Nisson et al. 2023, and in the case of the Raman-based counts for the 95-level brine, are dependent on cells displaying non-Brownian motion (suggesting they are active).

Edited Lines: 1387-1388

(3) Lines 746 – 771: It's not clear how microbial survival times were calculated. These should be described here.

Per our previous response to the comment 2c and on line 340 in the manuscript, this part of the methods has been expanded to clarify which values were drawn from the literature to determine microbial 'survival times' under the dosage/particle type conditions of our Moab Khotsong-specific model. This includes specification that each case considers a single-species starting population, and relies on the accumulated dosage (in Gy) at which each population is considered 'inactive', as drawn from the literature.

Edited Lines:1900-1928

Other minor comments:

Line 27: add a qualifier to mention the source of the radiation

We agree with the reviewer in that the sources of 'radiolytic enrichment' for the Moab Khotsong site could benefit from further clarification. As a result, the phrasing of this line was changed from 'highly radiogenic' to 'uranium-enriched' to highlight the driving source of radiation for this system. We believe this change of phrase pairs well with additional discussion of radiation sources at Moab Khotsong added to the Introduction in line with the Reviewer's comment 1b.

Lines edited: 33

Line 29-30: Define acronyms on first use, or use alternative term for abstract (e.g., 'spectroscopic and mass spectrometric approaches suggest...')

Abbreviations presented throughout the abstract and main text of the manuscript were revised so that their full definition is presented upon first appearance. If abbreviations appear in both the abstract and main text, they are now accompanied by a full definition of the term on their first appearance in both (abstract and main text). The first appearance of abbreviations in the main text have been edited to include a brief description of the technique and what we are trying to accomplish by employing the technique.

Edited Lines: 35-37, 338, 598, 631, 702

Line 53 – Greater diversity than what? Please clarify

We have added a brief clarification and context to this section to confirm we are discussing these shallower systems relative to deeper, longer-isolated fluids.

Edited Lines: 191-192

Lines 113 – 114: He authors might consider using the term 'stress' instead of thermodynamic constraints- not all stresses imposed by salinity, temperature, etc. will relate to thermodynamic constraints.

We agree with the reviewer that 'stress' is a more suitable term to describe the implications of high salinity, temperature, and radionuclide decay exposure on inhabiting microbial life. The term 'thermodynamic constraints' was changed accordingly to 'stress'.

Edited Lines: 263

Line 116: It's not clear what is being compared here – higher biomass than what? Please clarify

We appreciate the reviewer's suggestion for clarity in this section. We do have optical microscopy and Syto-9 based cell counts for cells found in the 95- and 101-level brines which were initially (and briefly) presented in Nisson et al., 2023. The brine cell counts are up to 4 orders of magnitude lower than cell counts of the 1200-level dolomite fluid. It is the brine cell counts we are referring to here as the relative values. We have added in these cell counts to this section so the comparison of 1200-level biomass to that of the Moab Khotsong brines is clear. It is extremely difficult to extract enough DNA for high quality sequencing from the brine fluids, and we believe a more thorough investigation of sequence-based evidence for brine

microbial life is better suited for a future and separate manuscript; as a result, throughout this manuscript we limit investigations of brine 'life' to an energetic perspective.

Edited Lines: 266

Line 126: In what sense are these settings both 'isolated' and 'global'? Please clarify and add a citation if necessary.

We have clarified accordingly and have also cited Warr et al. (2021) here, here, which specifically summarizes convincing subsurface fluid system residence times based on stable isotopic signatures of $\delta^2\text{H}$ and $\delta^{18}\text{O}$ (great indicating values of meteoric mixing, or a lack thereof) on a global scale to fully address this point by the reviewer.

Edited Lines: 323-324

Figure 1: d13C arguments in the Discussion would benefit from plotting on here expected ranges from various biotic/abiotic processes drawn from the literature.

We appreciate the suggestion from the reviewer and agree that the addition of d13C value ranges from relevant sources should be added for clarification and quick comparison to figure 1. We have elected to provide ranges for d13C of DOC and DIC associated with previous published values from the Witwatersrand Basin lithology and fracture fluid environments. By constraining presented ranges to Witwatersrand Basin-derived values, we aim to avoid site-specific reaction-rate and carbon source discrepancies in the interpretation of Moab Khotsong values. These 'ranges' have been added to the figure 1 in the form of double ended arrows spanning particular d13C values, or a single ended arrow if the range exceeds d13C axis values for figure 1 (in this case the full range is provided in the caption). These are included along with their corresponding references in an updated caption for this figure. Some clarifications were added to the d13C DOC/DIC discussion in the conclusion section to best fit with these new figure changes.

Edited Lines: Figure 1, 390-405, 859, 889

Line 194: Should be 'spectrum' (singular)

The term 'spectra' was changed to singular 'spectrum' in the text.

Edited Lines: 477

All figures/tables: I suggest moving these to after their first mention in the text

All figures/tables have been moved to follow their first mention in the main text of this manuscript, unless already in that position.

Edited figures: Figure 1, Figure 3, Figure 4, Figure 5, Figure 7, Figure 8; Table 1, Table 2.

Lines 483 – 486: It's not clear what setting you are describing here. Please clarify where you expect to find cryogenic brines in the subsurface.

We appreciate the reviewer's comment. Upon secondary review, it appears somewhat out of place and unnecessary and so we have decided to delete this sentence from this section of the discussion. All discussion of microorganisms in cryogenic settings are now introduced later in this section, where potential maintenance restrictions on life in the subsurface of Europa, Enceladus, and Ceres are mentioned.

Edited Lines: 1394-1453

Lines 514 – 515: It's not clear what setting on Europa and Enceladus the authors are describing here. The temperatures at 1m depth on Europa and Enceladus will be $\leq 100\text{k}$. There will be no liquid water and even volatiles like methane will be frozen.

We appreciate the reviewer's comment on the temperature discrepancy for these systems. We agree that the discussion for Europa and Enceladus is confusing as presented for icy near-surface regions. As a result, we have decided to focus the brief discussion of these icy moons to consider radiolytically generated species in their deep ocean regions. Although in several cases, the radionuclide concentrations are not well constrained for deeper rock interiors of these planets, there are estimated ranges which have been applied to previous habitability models considering production of radiolytic e- donors and acceptors (Bouquet et al., 2017; Altair et al., 2018) such as H₂ and sulfate. Also, these considerations focus on radionuclide-fueled radiolysis, which is similar to Moab Khotsong. For the discussion of Mars, we have expanded this to include ionizing radiation dosages at near-surface sites (primarily dominated by galactic ionizing radiation; $\sim 0.3\text{ Gy/yr}$ at 1 m depth (Dartnell et al., 2007)) and to include deeper, regolith decay-dominated radiolysis ($\sim 0.03\text{ Gy/yr}$ at 1m depth (Teodoro et al., 2018)) that may be able to support low biomass ($\leq 10^6\text{ cells/(kg rock)}$) communities of sulfate reducers (Tarnas et al., 2021)). Dosage estimates presented for the Martian subsurface are revised to reflect the terrain used in their associated models from the literature (e.g. wet heterogenous or dry homogenous regolith).

Edited Lines: 1385-1394

Lines 520: I suggest the authors consider implications for Ceres, where subsurface brines have been inferred.

We appreciate the authors suggestion for the inclusion of dwarf planet Ceres in our discussion of radiolytically active icy ocean worlds (newly added reference Bouquet et al., 2017 provides a look at radiolysis on Ceres alongside Europa and Enceladus). We have included Ceres in our brief discussion alongside Europa and Enceladus, but like these moons, we now clarify that radiolytic activity, and potentially radiolytic supported habitability, is likely restricted to deeper (km scale) brine oceans. These regions are suggested to receive input from radionuclide decay of rock particles proximal to the brine environment.

Edited Lines: 1394-1453

Line 725: Seems very high. Should this be mmol?

The concentration of mol/L is accurate. This relatively high concentration was used in the model to represent a large potential pool of labile substrates as the breakdown products from the larger, and likely more recalcitrant overall DOC pool. This does not present an issue, as the free energy flux (FEF) calculation includes consideration for the limiting substrate in each metabolic strategy; even if written into the model at 1 mmol/L concentration, hexose would not be the limiting reactant in any of the reactions in which it appears, and its concentration would not have any significant effect on the final FEF calculated value.

Reviewer #2 (Remarks to the Author):

The Nisson et al. manuscript on the ancient organic matter pool feeding the deep biosphere in brines of South African mine informs us about the potential habitability and possible microbial metabolisms on this extreme environment. The results show the importance of radiolytic reworking of ancient organic carbon pool of host rocks to microbial communities, in addition to already established radiolytically formed electron acceptors such as oxygen and nitrogen. Interestingly, the microbial metabolism this environment seems to best support is microaerophilic heterotrophy. Similar ideas have been previously suggested at least for the Fennoscandian deep biosphere. Thus, the study gives further confirmation about the importance of heterotrophy in deep terrestrial subsurface habitats. The abiotic water-rock interactions are the main source of organic matter that the low cell number -microbial communities are able to utilize, and according to energy flux calculations, again, heterotrophic metabolism combined with aerobic or nitrate reduction provides enough energy for the microbes to survive long time periods in the hypersaline and hot deep subsurface environment. Alongside, the authors also modelled the capacity of well-known radiotolerant and -resistant bacterial species to survive in these conditions, leading to estimate of survival up to 750000 years by the most radioresistant microbes.

I found the manuscript well-prepared, clear and concise, and results supported the conclusions. Some methods (EEMs and negative ion ESI 21 tesla FT-ICR MS) are somewhat unfamiliar to me, hampering the assessment of their relevancy, but the results are clearly explained and credible. I have only minor details that the authors might want to take into consideration, but I leave it up to authors to decide if they found these useful:

-adding a bit more detail on the discussion about heterotrophic metabolism in the deep biosphere. There are a few studies where this has been touched and this could be elaborated in the manuscript. In addition, astrobiological discussion on habitability has concentrated on serpentinization and autotrophy, but this manuscript shows another potential energy and carbon source for extraterrestrial microbes.

In response to this reviewer's comment, we have added a brief sentence touching on sedimentary organic matter-fueled deep fluid microbial systems in the discussion, but did not expand this further due to limited space. Additionally, the discussion of radiolysis influencing habitability in planetary environments has been expanded to discuss specific regions of the Martian subsurface, icy dwarf planet Ceres, and icy moons Europa and Enceladus, in line with suggestions from Reviewer #1.

Edited Lines: 1232-1240, 1394-1453

-adding some comparison about the time frame the microbial community could have survived in the Moab Khotsonq deep subsurface in contrast to other old fluids in Canada, SA and Fennoscandia.

We appreciate the reviewer suggestion to add some comparative survival timeframes for other global deep fluid systems. As a result, a few sentences have been added to the discussion presenting potential time windows over which microorganisms may have survived in

Precambrian Shield settings, including South Africa's Kaapvaal Craton, the Fennoscandian Shield, and for the Canadian Shield. All of these systems have minimum potential habitable fluid timescales as old as millions of years and are based on thermal constraints detailed in reference Drake et al., 2021.

Edited Lines: 1315-1379

- some more details on the lithology and fracturing of the rock would be great. Was Vaal Reef sample analyzed because it might be a potential source of the DOC pool in Moab? Is there any knowledge on the composition of shales surrounding the 95 and 101 sites in the likely more porous sandstone units? Could those also host a pool of organic matter?

We agree with the reviewer that additional detail would be helpful to further clarify the relationship between organics of the Vaal Reef zone and the lower shale regions. It has been proposed that there is a genetic relationship between carbon of the reef and Jeppetown shales in Gray et al., 1988. In this study, the authors suggest a past hydrothermal event in which refractory organic matter of the deeper shales migrated upward through hydrothermal induced fractures, contributing to the hydrocarbon content of the reef zone. We have added a brief discussion of this and included the reference to Gray et al. 1998 in the section of the discussion where the Vaal reef and shales are introduced.

Edited Lines: 857-864, 1502-1509

Best,

Lotta Purkamo, Geological Survey of Finland

Reviewer #3 (Remarks to the Author):

Summary

This is a very thorough and interesting study of carbon cycling and life in a deep subsurface ecosystem that has been isolated from the surface world for an extraordinarily long interval of time (>1 Ga). It provides novel insight into one of the most enigmatic ecosystems on Earth. Many of its results are noteworthy and will be significant to the field (see conclusions 1-3 below). With appropriate major revision, it will be very appropriate for publication in Nature Communications.

The primary conclusions are as follows:

1. EEMs, negative ion ESI 21 tesla FT-ICR MS, and amino acid analyses suggest the brines likely contain a large geologically "old" biotic carbon pool, derived from radiolytically oxidized kerogen-rich shales or reefs, that overshadow current microbial contribution.
2. The GC and isotopic analysis of volatiles reveal predominately n-alkanes (<C6) of methane and ethane, and C1-C3 hydrocarbon $\delta^2\text{H}$ and $\delta^{13}\text{C}$ signatures consistent with an abiotic origin.

3. These findings suggest that over the long subsurface isolation of the brine, water-rock processes control redox and C cycling, lending support for a low biomass, “slow biosphere” of micro-aerophilic and heterotrophic microbes supported by radiolytic oxidants.

These primary conclusions are generally well-supported. They constitute a substantial advance in understanding of deep subsurface life.

The evidence for the geologically ‘old’ biotic carbon pool and its origin appears solid. The ^{14}C signature of this carbon was initially surprising to me, but the reporting summary makes a reasonable case that the ^{14}C was created primarily, perhaps entirely, in situ (as suggested at the top of page 22).

The evidence for an abiotic origin of the n-alkanes (<C₆) of methane and ethane, and C₁-C₃ hydrocarbon $\delta^2\text{H}$ and $\delta^{13}\text{C}$ signatures also appears solid.

The evidence for low biomass requires more nuanced discussion, but the very low biomass estimates are likelier to be too high than too low. See below for details.

The inference of micro-aerophilic and heterotrophic microbes supported by radiolytic oxidants is reasonably well supported by the energies of reactions.

Primary concerns

My primary concerns are associated with the calculations and assumptions focused on microbial metabolism, maintenance, and cell densities. The cell density calculations require clearer and more thorough discussion in a revised manuscript. The “free energy flux” calculations and the cell maintenance calculations pose more intractable problems. These calculations are not central to the primary conclusions (1-3 above). As they’re presently calculated and presented, they significantly weaken the manuscript.

1. Cell density estimates are derived from aspartic acid concentrations (Table 1 and page 16). This derivation appears to implicitly assume that the aspartic acid is solely associated with intact cells, rather than with dead biomass or in dissolved form. This assumption should be explicitly acknowledged. Primary contribution from dead biomass would be consistent with the page-24 mention that “Cell density estimates based on aspartic acid concentration were 1-2 orders of magnitude greater than original Syto-9 or Raman-based cell counts for each sample (from reference 39). The abstract appears to implicitly acknowledge this point by giving a cell density range that is 2 to 5 orders of magnitude lower than the estimated densities in Table 1 (for Level 95 filtered and Level 1200, respectively). The difference between the Table 1/Results aspartic-acid-based estimates and the abstract estimates should be clearly explained by discussing this topic more thoroughly in the manuscript.

We appreciate the reviewer’s comment and call for clarification on the assumptions implicit to the cellular density calculations. We agree that the association to intact cells must be explicitly stated in the text. As a result, we have added a statement clarifying this assumption to

cellular density estimates as presented in the results, discussion, as well as the methods. Additionally, we have expanded the discussion on this point to acknowledge that the discrepancies between microscopic cell counts, and aspartic acid-based cell density estimates, may be due to contributions from dead biomass and/or a dissolved fraction. This can also be seen when looking to cell density estimates (that rely on the same intact cellular assumption) from the 95-level filtered fraction, in which most of the intact cells should have actually been retained on the filter (0.2 μm); despite using the same intact cell assumption for biomass estimates of this sample, it gives us a general idea that there is some aspartic acid contribution from an extracellular fraction. In the methods section, we expand upon the estimate calculation of biomass from aspartic acid concentration, as this calculation relies on literature values calculating the fraction of aspartic acid in bacterial cells from Moura et al. 2013.
Edited lines: 1073-1122, 1683-1689

2. It's surprising to me that racemization age estimates were not calculated for the aspartic acid enantiomeric ratios. Why was this not done? Would it strengthen or weaken the arguments of the manuscript?

While using enantiomeric ratios to determine racemization age estimates is a useful technique for systems with a well-recorded temperature history and sample geologic age, we do not meet both of those criteria with the Moab Khotsong brine system. Specifically, we do not have a tight constraint on the timings or volumes of any potential fluid infiltration that might have introduced a viable microbial community, and thus do not have any good constraint on the age of these fluid fractions. We know based on the temperature history of this system that the *in situ* community cannot be as old as the primary fluid body (dated at 1.2 Ga with noble gases (Warr et al., 2022)), since over this time the temperature exceeded that for microbial life (>150°C; ~350°C maximum). Additionally, the extent of potential radioracemization of aspartic acid is not known, and while radioracemization is not typically of concern relative to radiolytic degradation of amino acids (also a factor that would influence this system), there could be a small effect (Pavlov et al., 2022). As presented, we believe the addition of aspartic acid racemization age estimates would not add significantly to answering the major question imposed in this section of our manuscript (is there currently a significant contribution from *in situ* microbial life to the organics pool?) and if included, would require a much more thorough investigation into concerns of radiolytic effects on the amino acid pool, one which is outside the scope of this study and its intent. As a result, we elect not to present aspartic acid racemization age estimates, and instead focus on the D/L ratios of filtered and non-filtered brine fractions.

3. The "free energy flux" calculations appear very problematic. Based on the page-38 description, free energy flux is calculated from assumptions of cell radius, diffusivity to the cell center, and constant concentrations of the reactants, with no consideration of reactant flux rates to the ecosystem or product flux rates from the ecosystem. This calculation appears to assume that reactants for all the cells enter the environment and products from all the cells exit the environment as fast as the "limiting substrate" can diffuse the distance of the assumed cell radius. This is a huge assumption. It seems unlikely to be true since timescale of diffusion is directly related to the square of the distance, a cell radius is very small, and the ultimate sources and sinks of the reactants and products are probably far from cells. By assuming a fixed

“substrate” concentration and a fixed *in situ* free energy, the calculation also assumes that reactant and product concentrations are constant, although the reactions remove the reactants and produce the products. These assumptions can only be true if reactants and products are replaced and removed as rapidly as they are respectively used and created. Without quantitative consideration of the rates at which reactants enter and products leave the larger ecosystem, the cell-radius “free energy flux” calculations seem unlikely to capture the true flux of free energy to this >1-Ga ecosystem or the cells that inhabit it. I don’t see a way to resolve this problem without calculating energy fluxes in a way that explicitly accounts for how fast reactants enter the 95-level system as a whole.

We thank the reviewer for raising this and we have revised the manuscript to ensure we present the detail and limitations of the free energy flux model more explicitly in the Methods. To address the points raised here by the reviewer, there are three assumptions inherent to these energetic estimates that account for using a fixed substrate concentration and diffusion parameters:

1. We consider that the system has a ‘steady-state’ flow of nutrients.

This assumption is strong considering the old age of this system (1.2 Ga) and the lack of both stable isotopic and geochemical evidence to suggest significant mixing with other fluid systems. There should not be significant alterations in reactant availabilities over ‘short’ timescales, that would otherwise be captured over geologic time (e.g. at some point there may be much less pyrite to oxidize). The reactant/product concentrations utilized in the Geochemist’s Workbench script are based on current *in situ* measured values, and as a result, calculated *in situ* free energies for the 57 redox reactions reflect thermodynamic favorability of the present-day system.

Additionally, very similar free energy flux models have been published on a variety of fracture fluid systems within the Witwatersrand Basin deep subsurface (Onstott et al., 2004, 2006; Lau et al., 2014; Magnabosco et al., 2016). All these prior publications consider systems much younger than Moab Khotson, further strengthening this assumption in the present study.

2. We consider that microbial motility is insignificant relative to substrate diffusion in the system.

In this assumption, we assume microbes are not moving throughout the fluid and are instead associated with stationary mineral surfaces. This eliminates the need to account for variable microbial movement towards certain reactants. With this assumption one could imagine there may be microorganisms associated with a particularly radiogenically enriched zone of host rock, but this model ignores such heterogeneity.

3. Fluid advection is insignificant relative to substrate diffusion in the system.

This assumption is consistent with previously published noble gas and water isotope studies for this site (Warr et al 2021, 2022) indicating this is a diffusion dominated system, allowing reactant species to mobilize and reach stationary microorganisms in a reasonably uniform way. We acknowledge that these three points which are critical components of the model as stated above must be clearly indicated in the main text, and have added this to the improved version of the manuscript.

Edited Lines: 1786-1812

Why is 80% of the calculated free energy assumed to be available for metabolism? This is a high value.

We appreciate this comment from the reviewer and are keen to provide additional clarification here and in the manuscript regarding the microbial energetics model presented. Under standard cellular conditions (e.g. for a growing anaerobic cell with [ATP] 10 mM; [ADP] 1 mM, [P_i] 10 mM) we could expect that the energetic requirement to produce 1 mol of ATP would be 50 kJ/mol substrate (Thauer et al., 1977). We carry this assumption to aerobic microorganisms as it represents a maximum estimate of expenditure (this value of 50 kJ/mol may be lower, due to factors like greater phosphorus uptake under aerobic conditions (Kern-Jespersen and Henze, 1993; doi: [https://doi.org/10.1016/0043-1354\(93\)90171-D](https://doi.org/10.1016/0043-1354(93)90171-D)) but this is not well constrained). On top of this cost, we can quantify the amount of heat lost as ~20 kJ/mol, and add this to make a total metabolic cost of ~70 kJ/mol (Schink, 1997). We are arguing here however, that microorganisms in energy-limiting environments (which we believe to be a decent assumption for Moab Khotsong considering severe energy limitation imposed in other deep, long-isolated global brine systems (Lollar et al., 2019; Payler et al., 2019), and in high-temperature environments where excess metabolic heat loss may increase external temperatures beyond optimum (Frank et al., 2020)) result in microorganisms being able to reduce their heat loss to ~10 kJ/mol and leading to a total estimated metabolic cost for producing ATP/heat as ~60 kJ/mol (Schink, 1997). A similar estimate of 60 kJ/mol and a maximum metabolic efficiency of 80% has been used in free energy flux calculations for microbial communities in more mesophilic fluids of Beatrix gold mine in the Witwatersrand Basin (Lau et al., 2014). We do recognize this is a maximum efficiency estimate, and that real values may fall below this estimate. We have made sure to clarify this and associated assumptions in the methods section of the free energy flux model, and to clarify that this is a maximum energetic estimate anywhere this value appears in the main text.

Edited Lines: 713, 1226, 1786-1790

4. The maintenance energy calculations also appear very problematic. Maintenance energy is defined in very different ways by different authors and for different equations. Most of the equations used, including the one used in this manuscript, are not demonstrably relevant for calculating microbial growth or survival in natural ecosystems with very slow life. For a fairly thorough review of maintenance energy calculations and their associated problems (including for the equation used by this manuscript), see, e.g., P. van Bodegum, 2007 (Microbial Maintenance: A Critical Review on Its Quantification, *Microbial Ecology*, Vol. 53, No. 4, 513-523).

We appreciate this comment from the reviewer as a chance to add further context to the microbial maintenance energy model presented here. We certainly agree that throughout the literature there are presented a large variety of ways to describe microbial 'maintenance'. This is often categorized into (1) physiological maintenance (growth-related) and (2) non-growth-related strategies (e.g. defense against abiotic stress). Throughout the present study, we are explicitly focusing on the non-growth quantification of maintenance, in terms of Gibb's free energy requirement for a microbial cell.

The equation for temperature-associated maintenance included in the present study (Eq. 16 from Tjihuis et al., 1993) allows calculation of a maintenance energy (M_e) parameter dependent solely on environmental temperature. This equation is the result of correlations between M_e across different species types, different catabolic strategies, and anaerobic vs. aerobic strategies; this wide range of tested conditions implicitly examines a wide range of growth rates and establishes maintenance cost from temperature as a more significant cost. This is clearer in Eq. 17 of the paper, which establishes a way to quantify the total energy dissipation considering physiological and non-growth maintenance. Tjihuis proceeds to establish the non-growth term as always significant for temperatures $>50^\circ\text{C}$, despite differences in microbial growth rate. In this way we are presenting an estimate for the maintenance cost associated with non-growth factors that are likely very significant for microbial habitability, despite potential growth rates. In the case of Moab Khotson, these include high temperature (55°C) (and salinity near salt saturation) of this system. By evaluating only the non-growth related maintenance costs, in temperature defense and osmotic regulation, we are already able to see certain metabolisms inhibited by the abiotic stress of the environment (e.g. certain forms of methanogenesis, sulfate reduction in Table S2, Appendix I and II). The potential growth rate estimates for this system may be further explored in a future study devoted to maintenance estimates in deep brines, but they are outside the scope of the current manuscript. We have, however, included the explicit statement that we are focusing on non-growth (abiotic stress-associated) maintenance energy estimates. We have also altered the phrasing of the microbial maintenance sections to tone down the definitive nature of our statements.

Edited Lines: 1828, 1865-1869

The authors could most simply resolve concerns 3 and 4 (above) by deleting these topics (free energy fluxes and maintenance energy) from the manuscript. They are not central to the strong primary conclusions of the manuscript and they significantly weaken the manuscript in their present state. If the authors wish to retain these topics, they should either recalculate them using approaches that meet the fundamental concerns outlined above or explicitly describe the problems inherent in the present calculations and explicitly treat the calculated values as model predictions, rather than as definitive descriptions of free energy fluxes and maintenance energy in this extraordinary ecosystem.

Please see the above comments where we address in detail all assumptions, clarifications, and revisions made to microbial energetic estimates throughout this manuscript to reflect these points raised by the reviewer.

Secondary concerns

Why include free energy calculations for 57 total reactions when only 10 are discussed as favorable?

The decision to run a total of 57 different microbial reactions was made so that we could examine the energetic feasibility across a wide range of potential chemoheterotrophic, chemolithotrophic, and anaerobic/aerobic strategies and have the greatest idea of what types of metabolic strategies may be utilized in the brine. This contrasts with several previous studies employing the free energy flux estimation approach on non-brine Witwatersrand Basin fluids

(Lau et al., 2016; Magnabosco et al., 2016; Onstott et al., 2006 (citations included in manuscript)) which typically limit considerations to anaerobic, chemolithotrophic metabolisms. These previous fluids additionally did not have DOC concentrations comparable to the size of the Moab Khotsong brine DOC pools. For practicality we only present the top ten most favorable metabolisms in the main text to provide the reader a summary of which catabolic strategies are most likely thermodynamically supported in a brine-inhabiting microbial community, from a free energy flux perspective. The inclusion of the complete list in the supplemental material permits the reader to further examine individual strategies at their discretion. Our original submission included a sentence in the results to mention notable but unfavorable metabolisms that did not make the top 10. We also have now also added a sentence to the discussion to highlight how these unfavorable strategies are present in Witwatersrand Basin fluids, and other global brine systems, but not at Moab Khotsong.
Edited Lines: 1226-1230

Minor points:

Acronyms, such as EEMs and FT-ICR MS should not be used in the abstract or the manuscript without being defined when they're first used.

Abbreviations presented throughout the abstract and main text of the manuscript were revised so that their full definition is presented upon first appearance. If abbreviations appear in both the abstract and main text, they are now accompanied by a full definition of the term on their first appearance in both (abstract and main text). The first appearance of abbreviations in the main text have been edited to include a brief description of the technique and what we are trying to accomplish by employing the technique. This agrees with edits made in response to a similar comment from Reviewer #1.

Edited Lines: 35-37, 338, 598, 631, 702

Reviewer #1 (Remarks to the Author):

The revised manuscript addresses all of my concerns in a satisfactory manner. The new context and description of techniques, as well as the added nuance in the Discussion regarding microbial community structure and thermodynamic predictions of favourable metabolisms have, in my opinion, greatly strengthened the manuscript.

My only comment on the manuscript is as follows: when discussing the relevance of the Moab Khotsong brines to planetary brines, the authors claim "remarkable similarities" (Line 604). Given the preceding discussion of the lack of detailed knowledge of these planetary brines, it would be more accurate to begin this sentence: "The potential similarities between these prospective planetary sources and the radiolytically-driven Moab Khotsong brines...", or something to this effect.

Finally, I encourage the authors to double-check line numbers in their responses document before submission. These were mislabelled, meaning I had to go hunting for the relevant sections.

Reviewer #3 (Remarks to the Author):

As I said in my review of the original manuscript, "this is a very thorough and interesting study of carbon cycling and life in a deep subsurface ecosystem that has been isolated from the surface world for an extraordinarily long interval of time (>1 Ga). It provides novel insight into one of the most enigmatic ecosystems on Earth. Many of its results are noteworthy and will be significant to the field (see conclusions 1-3 below). With appropriate major revision, it will be very appropriate for publication in Nature Communications

The primary conclusions are as follows:

1. EEMs, negative ion ESI 21 tesla FT-ICR MS, and amino acid analyses suggest the brines likely contain a large geologically "old" biotic carbon pool, derived from radiolytically oxidized kerogen-rich shales or reefs, that overshadow current microbial contribution.
2. The GC and isotopic analysis of volatiles reveal predominately n-alkanes (<C6) of methane and ethane, and C1-C3 hydrocarbon $\delta^2\text{H}$ and $\delta^{13}\text{C}$ signatures consistent with an abiotic origin.
3. These findings suggest that over the long subsurface isolation of the brine, water-rock processes control redox and C cycling, lending support for a low biomass, "slow biosphere" of micro-aerophilic and heterotrophic microbes supported by radiolytic oxidants.

These primary conclusions are generally well-supported. They constitute a substantial advance in understanding of deep subsurface life."

My primary concerns were associated with the calculations and assumptions focused on free energy fluxes, maintenance, and cell densities. As I mentioned in the previous review, I believe these calculations and assumptions are not central to the primary conclusions listed above.

The revised manuscript appropriately addresses my concern regarding the cell density estimates. It does not meet my concerns regarding free energy fluxes and maintenance energy.

The assumptions described in the rebuttal (steady-state flow of "nutrients", limited microbial motility, and absence of advection) do not alleviate my fundamental concern about the calculation of free energy fluxes, which is that an energy of reaction can only be constant if reactants and products are transported to and from the reaction site as fast as the reaction occurs. These assumptions actually undercut the flux calculation because they imply that transport of reactants and products is diffusive within the 95 level. Because the timescale of diffusion depends on the square of diffusive distance, free energy flux must ultimately be set by the distance from the sources of reactants for the 95 level system and the distance to sinks of the reaction products, not by the cell radius used for the calculations in this manuscript. Since the cell radius is so small relative to possible distances from reactant sources, the approach used is likely to significantly overestimate free energy fluxes (possibly by many orders of magnitude).

It's important to also bear in mind that just because a calculated reaction yields energy does not mean that an organism will use the reaction. The organism may be incapable of using the reactant(s) under any circumstance, some factor other than available energy may limit reaction rates (such as availability of some key element), and/or the true energy of reaction may be lower than the calculated energy (if concentrations and/or other relevant properties are assumed & not measured).

In considering this topic (free energy fluxes), I looked at reference 38 because the revised manuscript says that "Calculations of ΔG and free energy flux relied on the inorganic and organic species composition previously determined for the 95 level brine [38] and in this study". The rebuttal similarly states that "The reactant/product concentrations utilized in the Geochemist's Workbench script are based on current in situ measured values...". Despite these statements, I could not find exact concentration data in this manuscript or reference 38 for the few reactants I looked for from the "Ten metabolic reactions with greatest free energy flux" (acetate, hexose, succinate). Where are these concentration data presented? If these calculations are completely based on measured concentrations, the concentrations (and all relevant environmental variables, such as pressure) must be clearly presented and easy to find so interested readers can find and assess them. If the calculations assume any concentrations and/or any other properties, those assumptions must be clearly stated also. Page 43 of the revised manuscript states that "A concentration of 1 mol/L was used for hexose (C₆H₁₂O₆), to represent a broad range of more labile organic compounds that could be a product of the brine DOC pool." This statement suggests that at least some of the reactant concentrations used for the free energy calculations and free energy flux calculations were assumed rather than measured. To the extent that this is true, neither the calculated free energy values nor any free energy fluxes derived from those values can be considered definitive for the 95-level system.

I appreciate the authors' elaboration of their perspective on maintenance energy and their revision of the manuscript with respect to this topic. However, criticisms of the approach used (from Tjihuis et al., 1993) (e.g., by van Bodegom, 2007) remain unaddressed. The results of the maintenance energy calculations are not mentioned in either the revised abstract or the revised discussion. Given this circumstance and the known difficulties associated with maintenance energy calculations, it's not clear to me why the topic is retained in the Results and Methods.

In closing, I think the revision improved many aspects of the manuscript. I also continue to think the manuscript's primary results are significant and worthy of publication in Nature Communications. However, I think the free energy flux calculations and the maintenance energy calculations continue to significantly weaken the manuscript. As I said in the first round of review, this issue can be addressed by either removing these topics from the manuscript or by significantly changing the manuscript's approach to them.

REVIEWER COMMENTS

Responses to reviewer comments are included below in blue text. Edits line numbers refer to lines in the manuscript version with tracked changes shown.

Reviewer #1 (Remarks to the Author):

The revised manuscript addresses all of my concerns in a satisfactory manner. The new context and description of techniques, as well as the added nuance in the Discussion regarding microbial community structure and thermodynamic predictions of favourable metabolisms have, in my opinion, greatly strengthened the manuscript.

My only comment on the manuscript is as follows: when discussing the relevance of the Moab Khotsong brines to planetary brines, the authors claim "remarkable similarities" (Line 604). Given the preceding discussion of the lack of detailed knowledge of these planetary brines, it would be more accurate to begin this sentence: "The potential similarities between these prospective planetary sources and the radiolytically-driven Moab Khotsong brines...", or something to this effect.

Finally, I encourage the authors to double-check line numbers in their responses document before submission. These were mislabelled, meaning I had to go hunting for the relevant sections.

We appreciate the reviewer's comment for highlighting the strong and potentially misleading wording in the original version of our manuscript. In response, we have replaced the initial part of this line with the reviewer's wording suggestion, to appropriately tone-down any claims of similarities between Moab Khotsong brines and those of other planetary systems.

Edited Lines (with tracked changes): 678-679

Reviewer #3 (Remarks to the Author):

As I said in my review of the original manuscript, "this is a very thorough and interesting study of carbon cycling and life in a deep subsurface ecosystem that has been isolated from the surface world for an extraordinarily long interval of time (>1 Ga). It provides novel insight into one of the most enigmatic ecosystems on Earth. Many of its results are noteworthy and will be significant to the field (see conclusions 1-3 below). With appropriate major revision, it will be very appropriate for publication in Nature Communications

The primary conclusions are as follows:

1. EEMs, negative ion ESI 21 tesla FT-ICR MS, and amino acid analyses suggest the brines likely contain a large geologically "old" biotic carbon pool, derived from radiolytically oxidized kerogen-rich shales or reefs, that overshadow current microbial contribution.
2. The GC and isotopic analysis of volatiles reveal predominately n-alkanes (<C6) of methane and ethane, and C1-C3 hydrocarbon $\delta^2\text{H}$ and $\delta^{13}\text{C}$ signatures consistent with an abiotic origin.
3. These findings suggest that over the long subsurface isolation of the brine, water-rock

processes control redox and C cycling, lending support for a low biomass, “slow biosphere” of micro-aerophilic and heterotrophic microbes supported by radiolytic oxidants.

These primary conclusions are generally well-supported. They constitute a substantial advance in understanding of deep subsurface life.”

My primary concerns were associated with the calculations and assumptions focused on free energy fluxes, maintenance, and cell densities. As I mentioned in the previous review, I believe these calculations and assumptions are not central to the primary conclusions listed above.

The revised manuscript appropriately addresses my concern regarding the cell density estimates. It does not meet my concerns regarding free energy fluxes and maintenance energy.

The assumptions described in the rebuttal (steady-state flow of “nutrients”, limited microbial motility, and absence of advection) do not alleviate my fundamental concern about the calculation of free energy fluxes, which is that an energy of reaction can only be constant if reactants and products are transported to and from the reaction site as fast as the reaction occurs. These assumptions actually undercut the flux calculation because they imply that transport of reactants and products is diffusive within the 95 level. Because the timescale of diffusion depends on the square of diffusive distance, free energy flux must ultimately be set by the distance from the sources of reactants for the 95 level system and the distance to sinks of the reaction products, not by the cell radius used for the calculations in this manuscript. Since the cell radius is so small relative to possible distances from reactant sources, the approach used is likely to significantly overestimate free energy fluxes (possibly by many orders of magnitude).

It's important to also bear in mind that just because a calculated reaction yields energy does not mean that an organism will use the reaction. The organism may be incapable of using the reactant(s) under any circumstance, some factor other than available energy may limit reaction rates (such as availability of some key element), and/or the true energy of reaction may be lower than the calculated energy (if concentrations and/or other relevant properties are assumed & not measured).

In considering this topic (free energy fluxes), I looked at reference 38 because the revised manuscript says that “Calculations of ΔG and free energy flux relied on the inorganic and organic species composition previously determined for the 95 level brine [38] and in this study”. The rebuttal similarly states that “The reactant/product concentrations utilized in the Geochemist’s Workbench script are based on current in situ measured values...”. Despite these statements, I could not find exact concentration data in this manuscript or reference 38 for the few reactants I looked for from the “Ten metabolic reactions with greatest free energy flux” (acetate, hexose, succinate). Where are these concentration data presented? If these calculations are completely based on measured concentrations, the concentrations (and all relevant environmental variables, such as pressure) must be clearly presented and easy to find so interested readers can find and assess them. If the calculations assume any concentrations and/or any other properties, those assumptions must be clearly stated also. Page 43 of the revised manuscript states that “A concentration of 1 mol/L was used for hexose (C₆H₁₂O₆), to represent a broad range of more labile organic compounds that could be a product of the brine DOC pool.”

This statement suggests that at least some of the reactant concentrations used for the free energy calculations and free energy flux calculations were assumed rather than measured. To the extent that this is true, neither the calculated free energy values nor any free energy fluxes derived from those values can be considered definitive for the 95-level system.

I appreciate the authors' elaboration of their perspective on maintenance energy and their revision of the manuscript with respect to this topic. However, criticisms of the approach used (from Tjihuis et al., 1993) (e.g., by van Bodegom, 2007) remain unaddressed. The results of the maintenance energy calculations are not mentioned in either the revised abstract or the revised discussion. Given this circumstance and the known difficulties associated with maintenance energy calculations, it's not clear to me why the topic is retained in the Results and Methods.

In closing, I think the revision improved many aspects of the manuscript. I also continue to think the manuscript's primary results are significant and worthy of publication in *Nature Communications*. However, I think the free energy flux calculations and the maintenance energy calculations continue to significantly weaken the manuscript. As I said in the first round of review, this issue can be addressed by either removing these topics from the manuscript or by significantly changing the manuscript's approach to them.

We understand the reviewer may not agree with all of our approaches to modeling microbial free energy flux and maintenance. Due to repeated concerns that these sections are weakening the manuscript, we have decided it is in the best interest to remove all sections from the main text that correspond to microbial free energy flux or maintenance energy considerations. The most significant deletions have included Table 2, all corresponding text in the results section, method sections "4.8 Biogeochemical Modeling of Supported Metabolisms", "4.9 Maintenance Energy for Temperature and Salinity", and "4.10 Microbial Survival Under Extended Radionuclide Decay", all paragraphs concerning these topics in the discussion, and Table S2 in the Appendix I. Any references to microbiology retained in the abstract, and main text sections of the manuscript, relate to results/discussion of current biotic contribution to the dissolved organic/inorganic carbon pools, or to the primary conclusion #3 (stated above by the reviewer), which includes suggestions that the carbon of the Moab Khotsong brine system may help support a low biomass community (supported by previous cell counts of Nisson et al., 2023 and aspartic acid estimated cell densities included in the current work). Additionally, the final paragraph in the discussion section in which radiolytic planetary brine environments are suggested as potential environments of increased habitability (based on the presence of low biomass in the radiolytic Moab Khotsong brines), was retained, as this paragraph does not directly include discussion of any microbial energetic calculations.

The title was also revised to reflect the habitability of the Moab Khotsong brines, but to exclude suggestion that habitability will be supported by quantitative models of microbial metabolism in this manuscript.

Edits Lines (with tracked changes): 1, 29, 38, 129, 131 (removed), 138, 142 (removed), Table 2 and corresponding results section (removed), Figure 7 and corresponding results section (removed), 426, 431-432, 554-558 (removed penultimate paragraphs of discussions section which included discussion of microbial energetics and radiolytic survivability), sections 4.8, 4.9, and 4.10 of the methods (removed), 1049, 1058-1059, Table S2 Appendix I (removed).